

# A new perleidid neopterygian fish from the Early Triassic (Dienerian, Induan) of South China, with a reassessment of the relationships of Perleidiformes

Zhiwei Yuan[1], Guang-Hui Xu[2,3], Xu Dai[1], Fengyu Wang[1], Xiaokang Liu[1], Enhao Jia[1], Luyi Miao[1] and Haijun Song[1]

[1] State Key Laboratory of Biogeology and Environmental Geology, School of Earth Sciences, China University of Geosciences, Wuhan, China
[2] Key Laboratory of Vertebrate Evolution and Human Origins of Chinese Academy of Sciences, Institute of Vertebrate Paleontology and Paleoanthropology, Chinese Academy of Sciences, Beijing, China
[3] CAS Center for Excellence in Life and Paleoenvironment, Beijing, China

Corresponding authors
Guang-Hui Xu,
xuguanghui@ivpp.ac.cn
Haijun Song, haijunsong@cug.edu.cn

## ABSTRACT

Neopterygii is the largest clade of ray-finned fishes, including Teleostei, Holostei, and their closely related fossil taxa. This clade was first documented in the Early Carboniferous and underwent rapid evolutionary radiation during the Early to Middle Triassic. This article describes a new perleidid neopterygian species, *Teffichthys elegans* sp. nov., based on 13 well-preserved specimens from the lower Daye Formation (Dienerian, Induan) in Guizhou, China. The new species documents one of the oldest perleidids, providing insights into the early diversification of this family. The results of a phylogenetic analysis recover *Teffichthys elegans* sp. nov. as the sister taxon to *Teffichthys madagascariensis* within the Perleididae. *T. elegans* sp. nov. shares three derived features of Perleididae: the length of the anteroventral margin of the dermohyal nearly half the length of the anterodorsal margin of the preopercle; the anteroventral margin of the preopercle nearly equal to the anterior margin of the subopercle in length; and the anteroventral margin of the preopercle one to two times as long as the anterodorsal margin of the preopercle. It possesses diagnostic features of *Teffichthys* but differs from *T. madagascariensis* by the following features: presence of three supraorbitals; six pairs of branchiostegal rays; relatively deep anterodorsal process of subopercle; absence of spine on the posterior margin of the jugal; and pterygial formula of D26/P14, A22, C36/T39-41.
The Perleidiformes are restricted to include only the Perleididae, and other previously alleged 'perleidiform' families (*e.g.*, Hydropessidae and Gabanellidae) are excluded to maintain the monophyly of the order. Similar to many other perleidids, *T. elegans* sp. nov. was likely a durophagous predator with dentition combining grasping and crushing morphologies. The new finding also may indicate a relatively complex trophic structure of the Early Triassic marine ecosystem in South China.

## INTRODUCTION

The Permian-Triassic mass extinction was the biggest biotic extinction in the Phanerozoic, wiping out more than 80–90% of marine species (*Erwin, 2006*; *Song et al., 2013*; *Fan et al., 2020*). Benthic animals, especially reef-building taxa, were severely decimated during the Permian-Triassic crisis (*Kiessling, 2010*). However, the diversity of nekton (cephalopods and fishes) was less impacted, probably because of their high motility (*Song, Wignall & Dunhill, 2018*). As an important component of the Modern Evolutionary Fauna, Neopterygii underwent an essential early evolution stage during the Early Triassic (*Sepkoski, 1981*; *Tintori et al., 2014*; *López-Arbarello & Sferco, 2018*; *Romano, 2021*).

Consisting of Teleostei, Holostei, and closely related taxa, Neopterygii occupies a predominant position in the composition of living ray-finned fishes, which has not always been the case throughout Earth's history (*Arratia, 1999*; *Friedman, 2015*; *Nelson, Grande & Wilson, 2016*; *López-Arbarello & Sferco, 2018*; *Xu, 2020b*). Since the earliest fossil recorded from the Mississippian (Early Carboniferous), Neopterygii maintained a very low diversity until the end-Permian (*Hurley et al., 2007*; *Near et al., 2012*; *Xu, Gao & Finarelli, 2014*). Neopterygii was less affected by the Permian-Triassic mass extinction (*Scheyer et al., 2014*; *Vázquez & Clapham, 2017*; *Smithwick & Stubbs, 2018*), and experienced rapid evolutionary radiation during the Early Triassic (*Xu & Gao, 2011*; *Tintori et al., 2014*; *Friedman, 2015*; *Romano et al., 2016a*).

To date, Early Triassic neopterygians have been identified on all continents except South America and Antarctica. Africa, particularly South Africa, Kenya, Tanzania, and Angola, is the main locality of Early Triassic freshwater neopterygians. They have been also recovered in Australia, China, Russia, France, and Germany (Fig. 1A). Over the last century, Early Triassic marine neopterygian localities have been uncovered in East Greenland, Madagascar, West Canada, and Spitsbergen. Recently, some marine neopterygians during this period have been reported from South China, India, and the Western USA (Fig. 1A).

During the Early Triassic, members of Neopterygii were mainly represented by stem-neopterygian taxa (*Romano et al., 2016b*), with only a few holosteans reported, namely *Angolaichthys* (*Teixeira, 1948*), *Paracentrophorus* (*Gardiner, 1960*), and *Tungusichthys* (*Berg, 1941*). The unambiguous representative genera of the Early Triassic Neopterygii include *Australosomus*, *Watsonulus*, *Helmolepis*, *Plesioperleidus*, and *Teffichthys*, all of which have been confirmed in previous cladistic analyses (*Stensiö, 1932*; *Lehman, 1952*; *Jin, Wang & Cai, 2003*; *Mutter, 2005*; *Tong et al., 2006*; *Marramà et al., 2017*). In addition, some neopterygians, represented by *Albertonia* and *Parasemionotus*, which are supposed to be parasemionotids but still need formal cladistic analyses, were known during the Early Triassic (*Lehman, 1952*; *Schaeffer & Mangus, 1976*).

Perleidiformes are a group of stem-neopterygian taxa that lived in both marine and freshwater environments in the Triassic. The paraphyly of the traditional Perleidiformes has been noticed in many previous phylogenetic studies (Table S1; *Xu, Gao & Coates, 2015a*; *Xu, Ma & Zhao, 2018*; *Wen et al., 2019*). Since being erected by *Berg (1937)*, 13 families have been assigned to this order (*Berg, 1940*; *Gardiner, 1967*; *Hutchinson, 1973*;

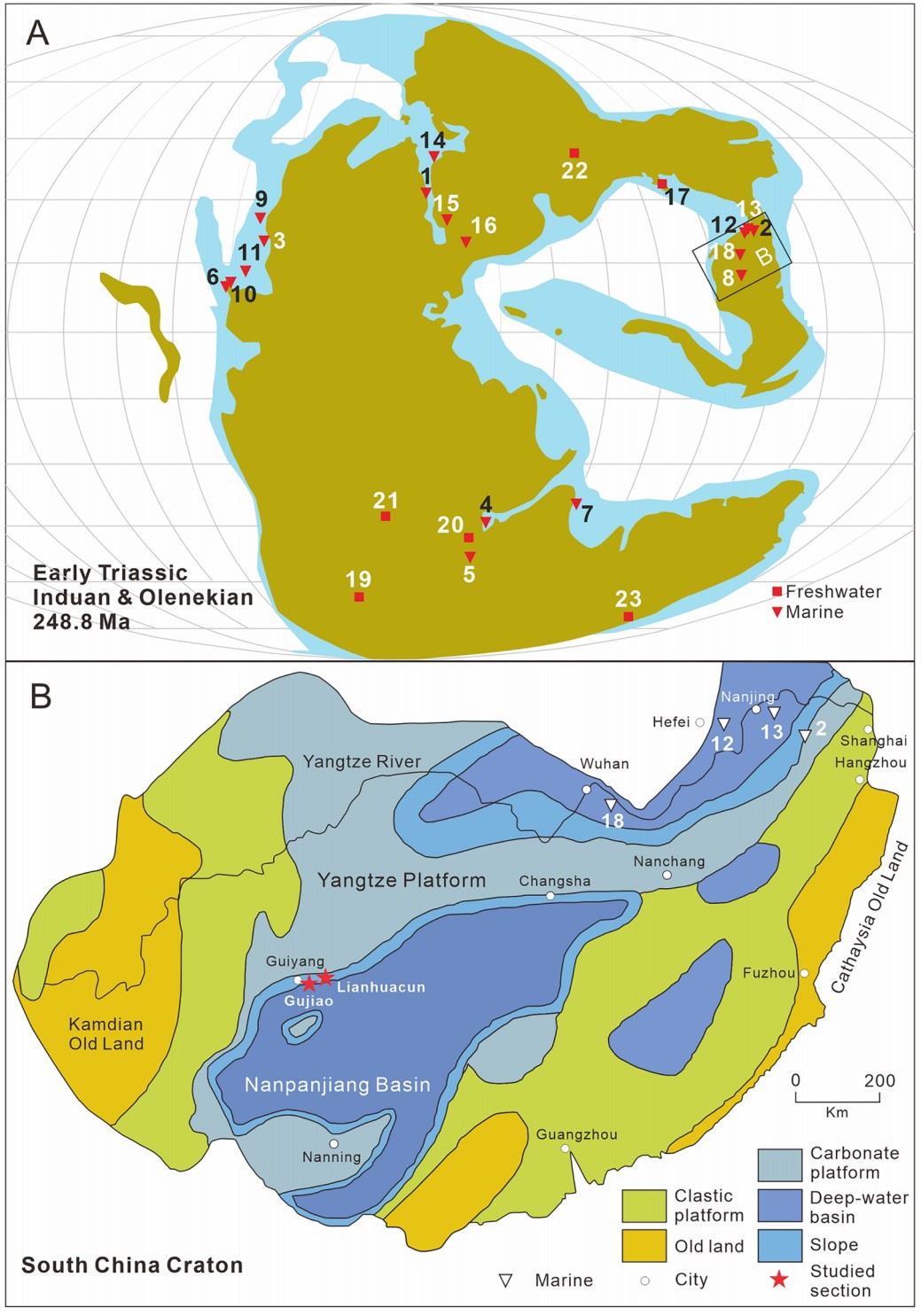

**Figure 1 Distribution of early triassic actinopterygii and studied location.** (A) Early Triassic Actinopterygii localities shown on the Paleogeographic Maps , modified from *Scotese (2014)*. (1) East Greenland (*e.g. Stensiö, 1932*; *Nielsen, 1942, 1949*); (2) Zhejiang, China (*Zhao & Lu, 2007*); (3) Alberta, Canada (*e.g. Lambe, 1916*; *Neuman, 2015*); (4) Northwestern Madagascar (*e.g. Lehman, 1952*; *Marramà et al., 2017*); (5) Southwest Madagascar (*e.g. Lehman et al., 1959*); (6) Elko County, Nevada, USA (*Romano et al., 2017*); (7) India, Spiti (*Romano et al., 2016b*) ; (8) Guizhou, China (this article); (9) British Columbia, Canada (*e.g. Schaeffer & Mangus, 1976*; *Mutter, 2005*); (10) Esmeralda County, Nevada, USA

**Figure 1 (continued)**
(*Romano et al., 2019*); (11) Idaho, USA (*Romano et al., 2012*); (12) Jiangsu, China (*e.g. Qian, Zhu & Zhao, 1997*; *Qiu et al., 2019*); (13) Anhui, China (*e.g. Tong et al., 2006*, *Sun et al., 2013*); (14) Spitsbergen (*e.g. Woodward, 1912*; *Stensiö, 1921*); (15) Poland (*Frech, 1903*–1908); (16), Germany and France boundary (*Gall, Grauvogel & Lehman, 1974*); (17) Gansu, China (*Xu, Gao & Coates, 2015a*); (18) Hubei, China (*Su & Li, 1983*); (19), South Africa (*e.g. Brough, 1931*; *Hutchinson, 1973*); (20) Tanzania (*Haughton, 1936*); (21) Angola (*e.g. Antunes et al., 1990*; *Murray, 2000*); (22) Siberia, Russia (*Sytchevskaya, 1999*); (23) Tasmania, Australia (*Dziewa, 1980*). Griensbachian: 1–2; Dienerian: 1, 3–8; Smithian: 9–14; Spathian: 11, 13–16; Early Triassic (stage indet.): 17–23. Neopterygii localities: 1–9, 12–19, 21–23. B. Map showing the fossil locality of *Teffichthys elegans* sp. nov. and Actinopterygii from South China, modified from *Feng, Bao & Liu (1997)*. 11 Actinopterygii from South China: 2, *Paraperleidus* from Zhejiang, China (*Zhao & Lu, 2007*); 13, *Plesioperleidus, Lepidotes, Stensionotus, Jurongia, Qingshania, Suius, Peia* from Jiangsu, China (*Qian, Zhu & Zhao, 1997*; *Liu et al., 2002*; *Jin, Wang & Cai, 2003*; *Li, 2009*; *Qiu et al., 2019*); 12, *Plesioperleidus, Chaohuperleidus, Jurongia, Qingshania, Suius* from Anhui, China (*Su, 1981*; *Tong et al., 2006*; *Sun et al., 2013*); 18, *Plesioperleidus* from Hubei, China (*Su & Li, 1983*).

*Bürgin, 1992*; *Tintori & Lombardo, 1996*; *Lombardo & Tintori, 2004*; *López-Arbarello & Zavattieri, 2008*; *Sun et al., 2012*; *Tintori, Lombardo & Kustatscher, 2016*). Among them, nine families were moved to other orders or abandoned: Luganoiidae and Fuyuanperleididae were moved to Luganoiiformes, Cleithrolepidae and Polzbergiidae were moved to Polzbergiiformes, Platysiagidae was placed within its own order Platysiagiformes, Pseudobeaconiidae was included within Louwoichthyiformes, and Teleopterinidae was moved to Amphicentriformes. Aetheodontidae was abandoned as the type and only known genus of this family assigned to Perleididae. Habroichthyidae and Colobodontidae were excluded from this order (*Griffith, 1977*; *Bürgin, 1992*; *López-Arbarello & Zavattieri, 2008*; *Van Der Laan, 2018*; *Xu, 2020b*). Consequently, Perleidiformes includes three families: Perleididae, Hydropessidae, and Gabanellidae. The latter two families only include a single genus each: *Hydropessum* Broom, 1913 from Hydropessidae *Hutchinson, 1973* and *Gabanellia* *Tintori & Lombardo, 1996* from Gabanellidae *Tintori & Lombardo, 1996* (*Hutchinson, 1973*; *Tintori & Lombardo, 1996*).

Perleididae is known from many Triassic fish localities, and it has a complicated taxonomic history. More than 30 genera have been grouped in this family. Nevertheless, many of them were revised later. At present, 16 genera are currently remained within Perleididae: *Meidiichthys, Perleidus, Manlietta, Procheirichthys, Plesioperleidus, Aetheodontus, Meridensia, Alvinia, Eoperleidus, Megaperleidus, Endennia, Paraperleidus, Diandongperleidus, Luopingperleidus, Chaohuperleidus,* and *Moradebrichthys* (*Brough, 1931*; *Wade, 1935*; *Su & Li, 1983*; *Bürgin, 1992*; *Sytchevskaya, 1999*; *Lombardo & Brambillasca, 2005*; *Zhao & Lu, 2007*; *Geng et al., 2012*; *Sun et al., 2013*; *Cartanyà et al., 2019*). It should be noted that the relationships among most of these genera still require further strict phylogenetic analysis.

The taxonomy of Perleididae in the Early Triassic is controversial. Apart from *Chaohuperleidus*, all perleidids in the Early Triassic have been historically placed in the genus *Perleidus* (*Stensiö, 1921*; *Stensiö, 1932*; *Piveteau, 1934*; *Lehman, 1952*; *Su, 1981*). *Lombardo (2001)* proposed that all current Early Triassic *Perleidus*-like fishes should be excluded from *Perleidus* due to the absence of epaxial rays. Later, *Marramà et al. (2017)*

erected a new genus *Teffichthys*, and proposed that, except *Plesioperleidus*, all other Early Triassic *Perleidus*-like fishes should be included in *Teffichthys*.

Abundant Early Triassic Neopterygii have been found in South China over the last 40 years, providing essential material to study the early evolution of Neopterygii and the origin of the modern evolution of fish fauna. Hitherto, 11 Neopterygii genera have been described from the Early Triassic in China (*Su, 1981*; *Su & Li, 1983*; *Qian, Zhu & Zhao, 1997*; *Liu et al., 2002*; *Jin, Wang & Cai, 2003*; *Tong et al., 2006*; *Li, 2009*; *Sun et al., 2013*; *Qiu et al., 2019*). They were mainly identified from Jurong, Jiangsu Province and Chaohu, Anhui Province, including seven and five genera, respectively (Fig. 1B). Interestingly, except for *Paraperleidus* from the Griensbachian (Early Induan), all the Early Triassic Neopterygii in China came from the Olenekian.

Here we describe a new perleidid species based on 13 specimens discovered from the Lower Triassic Daye Formation at two sections (Gujiao and Lianhuacun) near Guiyang, Guizhou Province, China (Fig. 1B). A calcareous concretion and 12 flat specimens from black shales are described in this study. The fossil beds were deposited in a basin or basin-margin environment (*Dai et al., 2019*). During the Early Triassic, Guiyang was located on the southern edge of the Yangtze Platform. In addition to the new species, other bony fishes (*e.g.*, coelacanths, parasemionotids) and invertebrates (*e.g.*, bivalves, ammonoids) were also discovered in the fossil beds. The age of the fossil beds is dated as Dienerian (Induan, Early Triassic), based on ammonoid and conodont biostratigraphy (*Qin, Yan & Xiong, 1993*; *Mu et al., 2007*; *Brüehwiler et al., 2008*; *Dai et al., 2019*).

## MATERIALS AND METHODS

All materials were stored at the YiFu Museum of China University of Geoscience, Wuhan (CUGM). The specimens were prepared using air chisels under a microscope. A three-dimensionally preserved skull of the holotype (CUGM K2-E2601) was prepared from both sides. A µCT scan was performed while useless information was obtained due to the internal mineralization. The specimens were photographed using a Canon 70D camera with a Micro EF lens with a focal length of 100 mm and f/2.8 aperture. Microscopic images were taken under a Leica S8 APO stereomicroscope. Illustrations were drawn using Adobe Photoshop CS6 and Coreldraw X7 software. The relative position of the fins and scale counts were expressed following *Westoll (1944)*. The traditional actinopterygian terminology (*e.g.*, *Gardiner & Schaeffer, 1989*; *Bürgin, 1992*; *Grande & Bemis, 1998*) was followed. The cheek bone located between the lateral edge of the dermopterotic and the anterodorsal edge of the preopercle is termed as a spiracular, following the nomenclature in some previous studies (*e.g.*, *Gardiner & Schaeffer, 1989*; *Bürgin, 1992*; *Marramà et al., 2017*), although it is also termed as a suborbital by others (*e.g.*, *Schaeffer & McDonald, 1978*; *Sun et al., 2012*).

To explore the phylogeny of Perleididae, a cladistic analysis was conducted based on materials of *Teffichthys elegans* sp. nov., and direct observations on materials of *Paraperleidus* (ZMNHM M1401) and *Plesioperleidus* (ZMNHM M8499, 8500; CUGM J2203a; IVPP V5343). In addition, we added *Moradebrichthys* and *Chaohuperleidus*, both of which are well preserved and have been placed within Perleididae (*Sun et al., 2013*;

*Cartanyà et al., 2019*; *Dai et al., 2021*). In order to explore the positions of Hydropessidae and Gabanellidae, we also added *Hydropessum* and *Gabanellia* based on literature data (*Hutchinson, 1973*; *Tintori & Lombardo, 1996*). The phylogenetic analysis was based on an updated dataset revised from the data matrix of *Xu (2021)*, containing 142 characters coded for 66 taxa. All characters were unordered and unweighted, and the basal actinopterygian *Moythomasia durgaringa* was selected as the outgroup taxon. Phylogenetic analyses were performed using the heuristic search algorithm (gaps treated as missing data, 500 random addition sequence replicates, tree bisection-reconnection (TBR) branch-swapping, with five trees held at each step and multiple trees saved) in PAUP* 4.0a169 (*Swofford, 2003*).

Anatomical abbreviations: ang, angular bone; ao, antorbital bone; bf, basal fulcrum; br, branchiostegal rays; cl, cleithrum; cla, clavicle; den, dentary; dh, dermohyal; dsp, dermosphenotic; exs, extrascapular; ff, fringing fulcrum; fr, frontal bone; ju, jugal; hm, hyomandibular; la, lacrimal bone; lg, lateral gular; mgu, medial gular; mx, maxilla; na, nasal; op, opercle; pa, parietal; pas, parasphenoid; pcl, postcleithrum; pmx, premaxilla; pop, preopercle; prr, procurrent ray; pr, principal ray; psp, postspiracular bone; pt, posttemporal; ro, rostral; sc, scale; scl, supracleithrum; so, suborbital; sop, subopercle; sp, spiracle; sr, sclerotic bone; su, supraorbital bone; s-ju, spiny process along the dorsal margin of the jugal.

The electronic version of this article in Portable Document Format (PDF) will represent a published work according to the International Commission on Zoological Nomenclature (ICZN), and hence the new names contained in the electronic version are effectively published under that Code from the electronic edition alone. This published work and the nomenclatural acts it contains have been registered in ZooBank, the online registration system for the ICZN. The ZooBank Life Science Identifiers (LSIDs) can be resolved and the associated information viewed through any standard web browser by appending the LSID to the prefix http://zoobank.org/. The LSID for this publication is: urn:lsid:zoobank.org:pub:D32FE268-726E-47C6-A2D9-F669836F3424. The online version of this work is archived and available from the following digital repositories: PeerJ, PubMed Central and CLOCKSS.

# RESULTS

## Systematic paleontology

Infraclass Actinopteri Cope, 1871
Superdivision Neopterygii Regan, 1923
Order Perleidiformes *Berg, 1937*
Family Perleididae *Brough, 1931*
Genus *Teffichthys Marramà et al., 2017*
Species *Teffichthys elegans* sp. nov.
LSID urn:lsid:zoobank.org:act:7AD93E0E-25F6-415E-8BBD-43872084F3D8
(Figs. 2–9).

**Etymology.** The specific epithet 'elegans' means elegant.

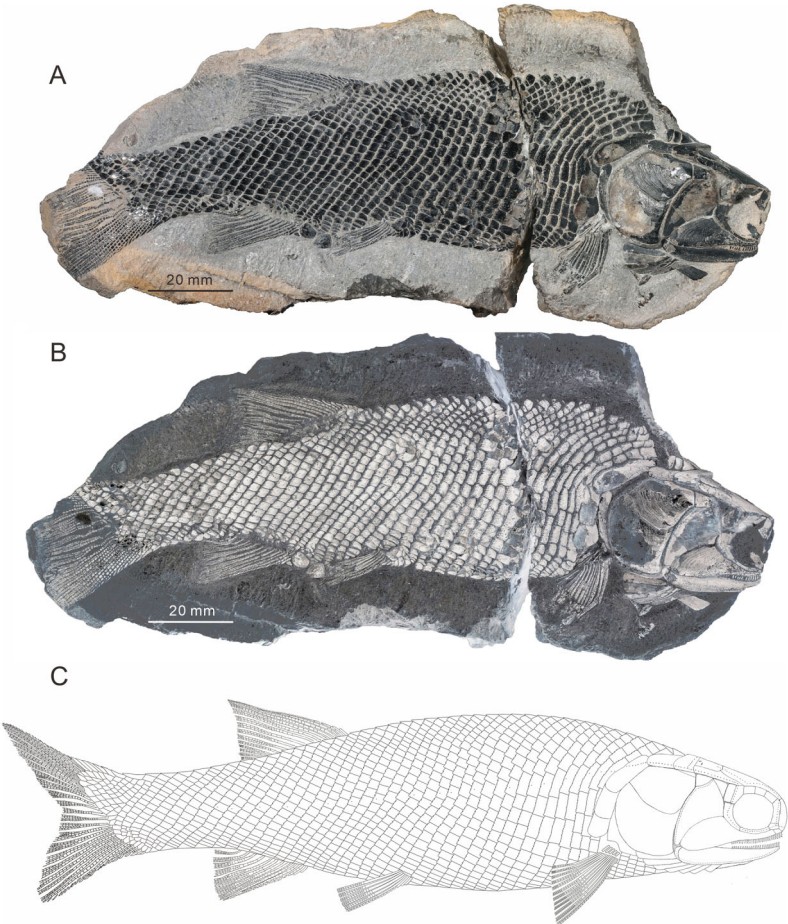

**Figure 2** **Holotype and the reconstruction of *Teffichthys elegans* sp. nov.** (A and B) CUGM K2-E2601, holotype from the upper part of Daye Formation at Lianhuacun section. (A) Photo of the specimen. (B) Invert counterpart of the specimen. (C) Reconstruction.

**Holotype.** CUGM K2-E2601. A specimen preserved in a calcareous concretion, with the snout region and the upper lobe of the caudal fin missing.

**Paratype.** CUGM K2-E2602–2604.

**Referred specimens.** CUGM K2-E2605–2613.

**Type locality and horizon.** Lianhuacun Village, Guiding County and Gujiao Village, Longli County of Guizhou Province, China; Daye Formation; Dienerian, Early Triassic.

**Diagnosis.** A new species of *Teffichthys* distinguished from the type species of this genus by the following features (autapomorphies identified with an asterisk): presence of three supraorbitals; six pairs of branchiostegal rays; relatively deep anterodorsal process of subopercle; absence of spine on posterior margin of jugal; and pterygial formula of D26/P14, A22, C36/T39–41 (*).

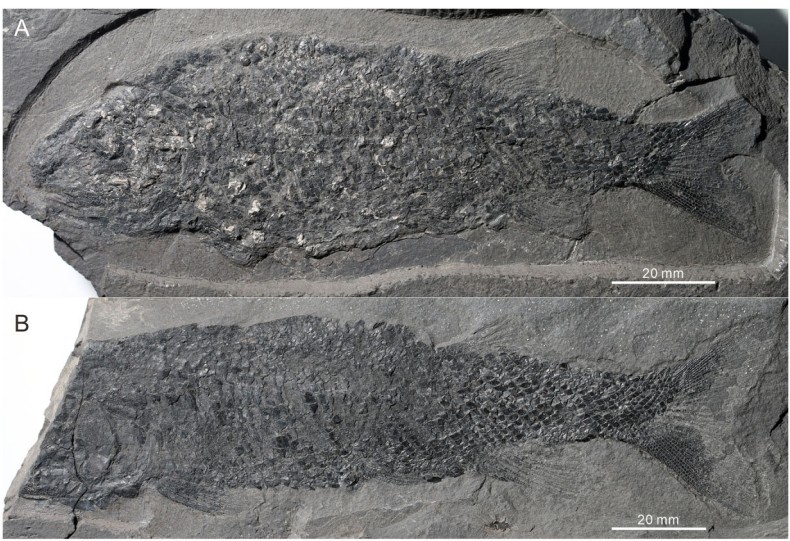

**Figure 3 Two paratypes of *Teffichthys elegans* sp. nov.** (A) CUGM K2-E2602. (B) CUGM K2-E2603. Both of them from the late Dienerian at Gujiao section.

## Description

**General morphology and size.** *Teffichthys elegans* sp. nov. has a blunt snout, an elongate and fusiform body outline, and a forked, nearly equilobated caudal fin (Figs. 2 and 3). The dorsal fin is larger than the anal fin, and it inserts slightly posterior to the origins of the pelvic fins. It has a standard length (SL; the length from the tip of the snout to the posterior extremity of the caudal peduncle) of 125–148 mm. The orbital region is relatively large, and its length accounts for 7.7–9.5% of the SL. The maximum depth of the body is located at the midway between the pectoral and dorsal fins, ranging from 37 mm to 51 mm. Head length and body depth accounted for 21.6–24.2% and 27.8–34.5% of the SL, respectively. The general body form is reconstructed based on the holotype and CUGM K2-E2602–2609 (Figs. 2–8).

**Snout.** A median rostral, the nasals, and the antorbitals are discernable from the snout region, and the premaxilla is poorly known because of incomplete preservation.

The rostral is the largest element of the snout region, being half of the length of the frontal (Figs. 5B–5D). It is broad and shield-like, reaching its maximum width lower to the boundary between the nasal and antorbital, then it becomes narrower up and down. There are no teeth on the rostral, excluding the possibility that it contributes to the oral margin. A distinct notch for the anterior nostril is present at the middle level of the lateral margin of the rostral (Fig. 5B). The surface of this bone is ornamented with tubercles (Figs. 5B–5D), and the ethmoid sensory canal is hard to identify.

The nasals are dorsoventrally elongated, shorter than the rostral in length (Figs. 5B–5D). It tapers dorsally and contacts the frontal and supraorbital dorsally, the antorbital ventrally, and the rostral medially. Similar to the rostral, the nasal is also ornamented with small tubercles on its external surface. A short anterior portion of the supraorbital sensory canal is present in the dorsal portion of the nasal (Fig. 5B).

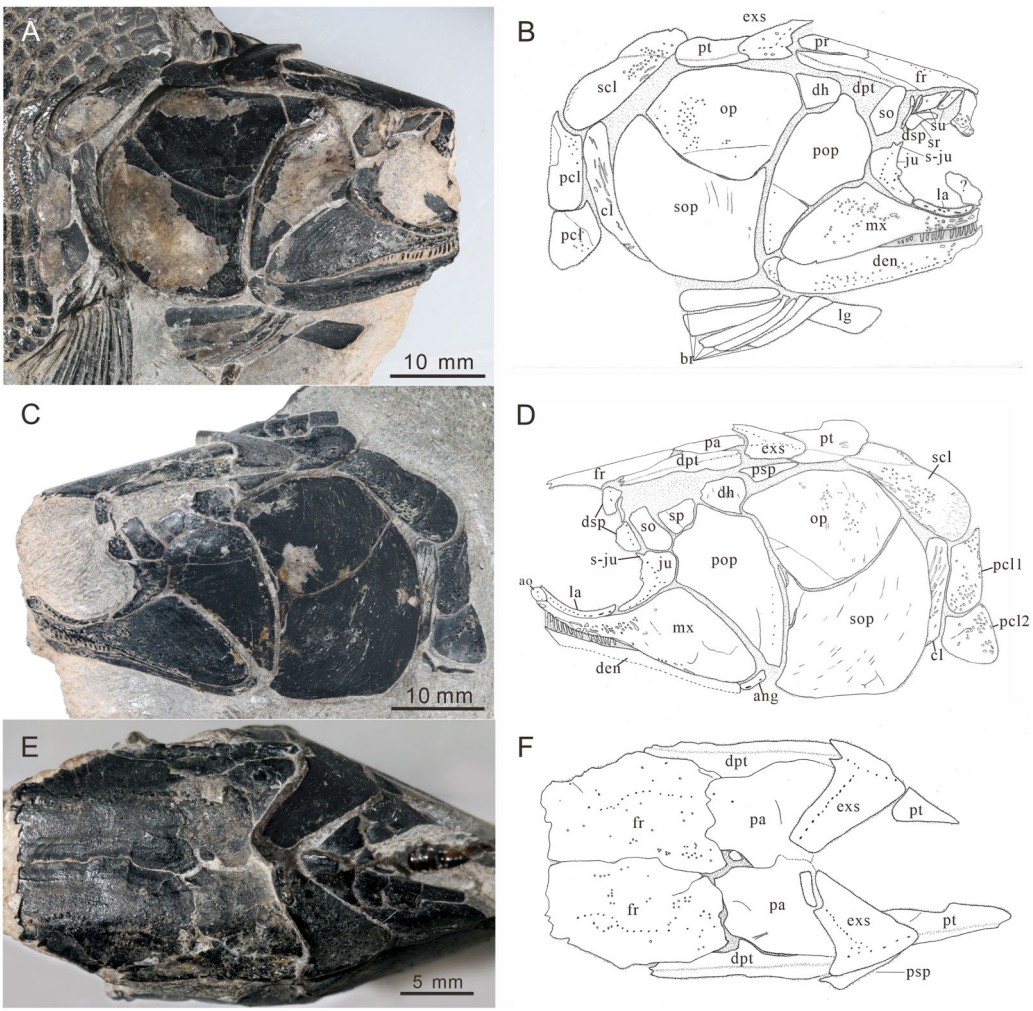

**Figure 4 Skull and pectoral girdle of CUGM K2-E2601, holotype from the upper part of Daye Formation at Lianhuacun section.** (A) Right view photograph. (B) Right view line-drawing. (C) Left side photograph. (D) Left side line-drawing. (E) Skull roof photograph. (F) Skull roof line-drawing.

The antorbital is rectangular, and is twice as wide as deep (Figs. 5B–5D). It contacts the nasal dorsally, the rostral anteriorly, the maxilla ventrally, and the lacrimal posteriorly. The posterior edge of the antorbital forms a part of the anterior margin of the orbit. It also bears dense small round tubercles on the surface. The infraorbital canal and ethmoid sensory canal meet on the upper portion of the antorbital (Fig. 5B).

The premaxilla is small, bearing more than two conical teeth (Fig. 6B). It contacts the rostral and antorbital dorsally and the maxilla posteriorly.

**Skull roof.** The skull roof is composed of a pair of frontals, parietals, dermopterotics, and extrascapulars.

The frontals are broad and trapezoidal in shape, slightly tapering anteriorly (Figs. 4E, 4F, 5C and 5D). Each frontal contacts its counterpart medially along a zigzag suture, and the parietal posteriorly with a wavy posterior suture. The short anterior part of the

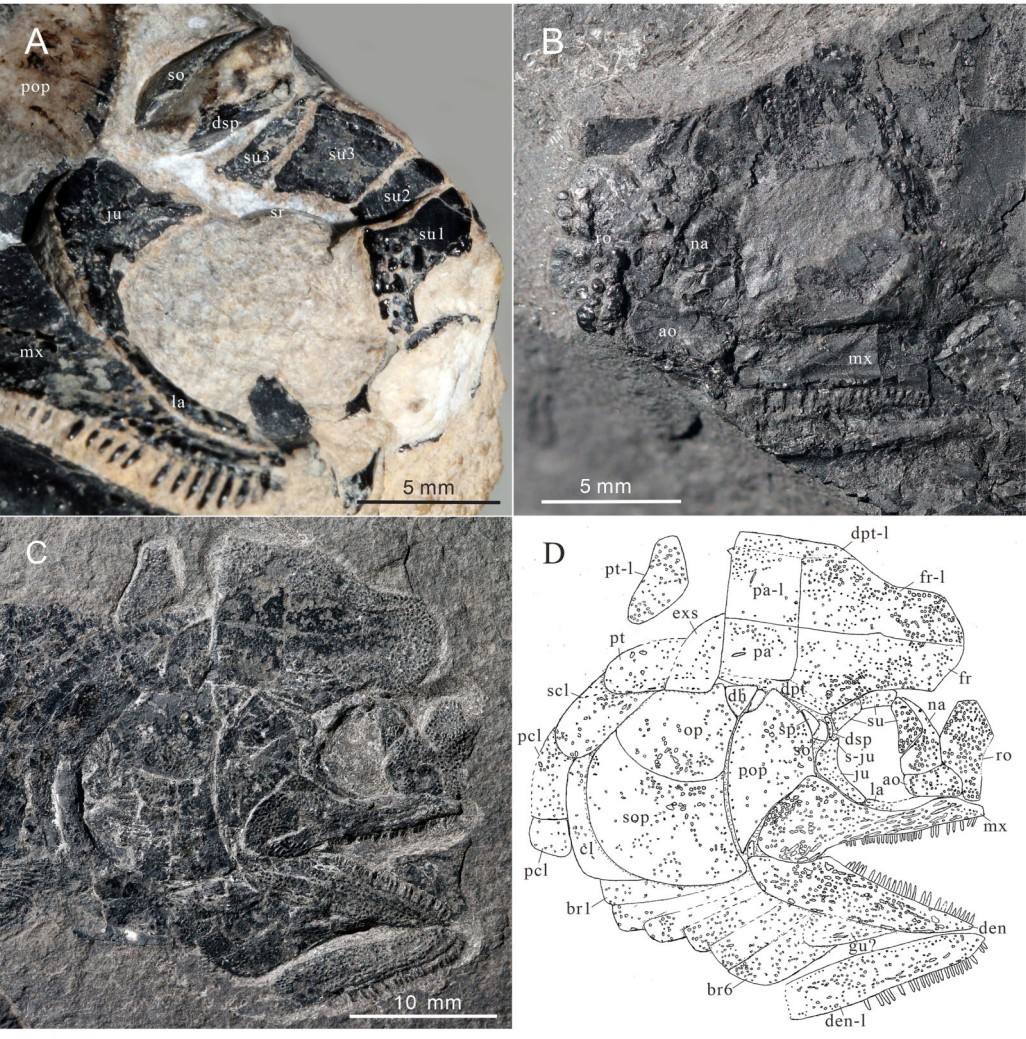

**Figure 5 Skull and pectoral girdle of *Teffichthys elegans* sp. nov.** (A) CUGM K2-E2601, orbital region. (B) CUGM K2-E2605, snout region. (C and D) CUGM K2-E2604; (C) skull and pectoral girdle, (D) line-drawing. All of them from the late Dienerian at Gujiao section.

frontals is missing in the holotype (Figs. 4E and 4F), while from specimen CUGM K2-E2604, it is unambiguous that the anterior region of the frontal contacts its counterpart medially (Figs. 5C and 5D). The length of the frontal is approximately twice the length of the parietal. The frontal reaches its greatest width at the level of the posterior border of the orbit, and slightly narrows towards the parietals.

The parietals are relatively small, rectangular in shape, and slightly broader than long (Figs. 4E and 4F). The median suture between the parietals is slightly curved. Each parietal contacts the dermopterotic laterally and the extrascapular posteriorly.

The supraorbital sensory canal runs longitudinally through the frontals along a sigmoid line, enters the parietal, then extends backwards for a short distance in this bone (Figs. 4, 5C and 5D). A middle pit-line can be recognized on the left parietal; the posterior pit-line on the right parietal is also discernable (Figs. 5C and 5D).

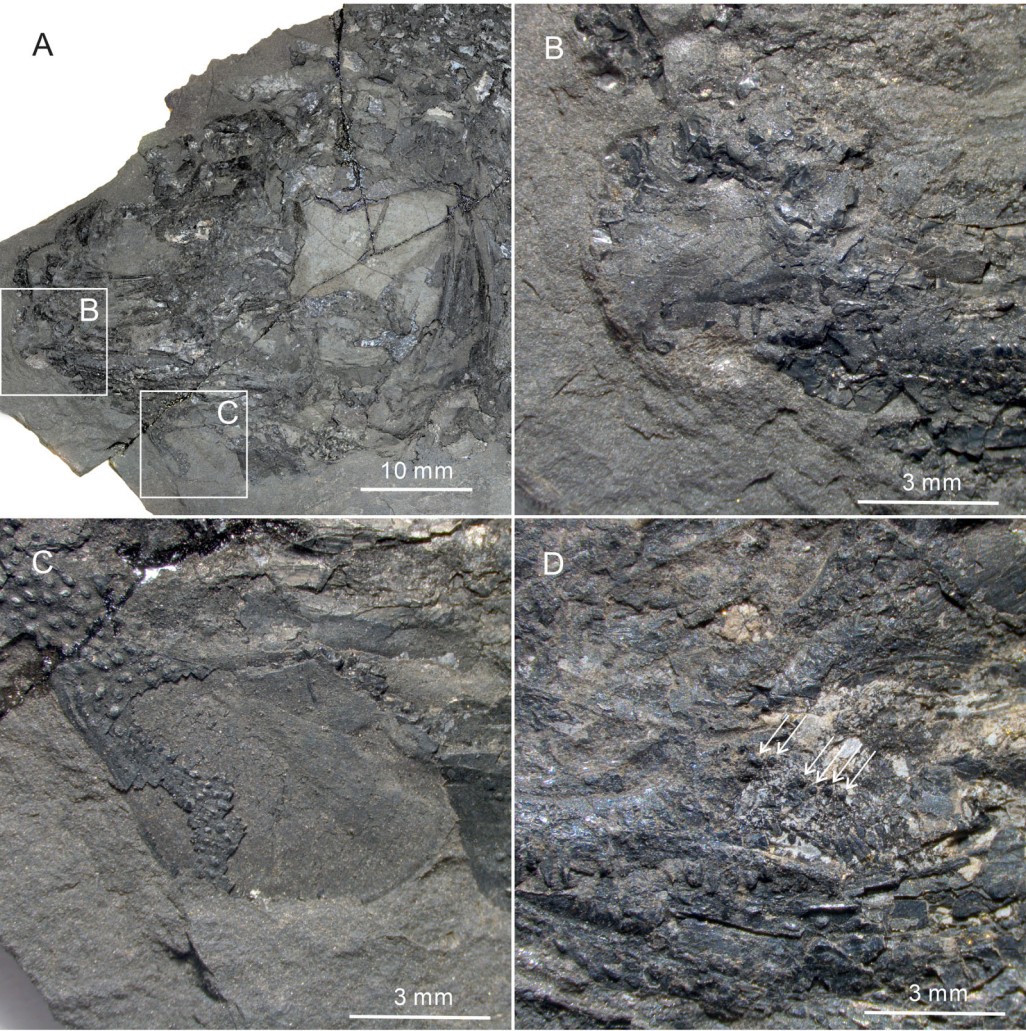

**Figure 6 Gular, premaxilla and molariform teeth of *Teffichthys elegans* sp. nov.** (A–C) CUGM K2-E2607; (A) head, (B) premaxilla, (C) middle gular. (D) CUGM K2-E2602, molariform teeth, indicated by arrows.

The dermopterotics are anteroposteriorly elongated, nearly twice the length of the parietal (Figs. 4A–4F). They contact the posterior portion of the frontal and the full length of the parietal medially. The supratemporal sensory canal runs through the dermopterotic longitudinally, and posteriorly enters the extrascapular through a notch at its anterior margin.

The extrascapulars are subtriangular in shape (Figs. 4A–4F). Each extrascapular tapers medially and contacts its counterpart medially, the parietal and dermopterotic anteriorly, and the posttemporal posteriorly. The supratemporal commissure runs through the anterior portions of both extrascapulars as is indicated by a series of small tubercles.

All these skull roof bones are ornamented with dense tubercles, well preserved on the specimen CUGM K2-E2604. Some ridges can be detected on the surface of the extrascapulars, which are usually parallel to the bone margin.

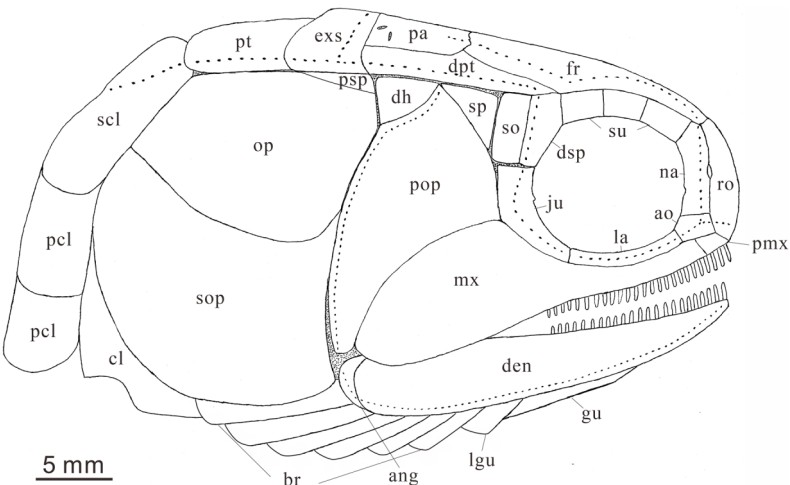

**Figure 7** Reconstruction of skull and pectoral girdle (without fin) of *Teffichthys elegans* sp. nov. from the late Dienerian (Induan, Early Triassic) at Gujiao and Lianhuacun section.

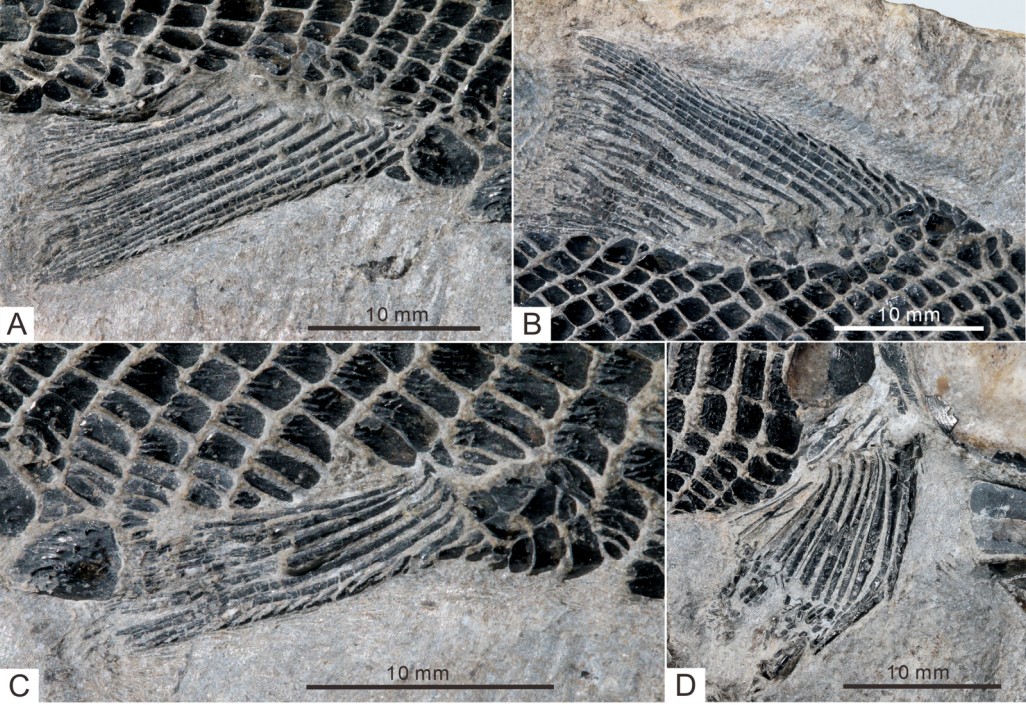

**Figure 8** Fins of *Teffichthys elegans* sp. nov. of CUGM K2-E2601, holotype from the upper part of Daye Formation at Lianhuacun section. (A) Anal fin. (B) Dorsal fin. (C) Right pelvic fin; (D) right pectoral fin.

**Circumorbital bones.** There are three rectangular to trapezoidal supraorbitals, contacting the frontal medially, the nasal anteriorly, and the dermosphenotic posteriorly (Figs. 4A, 4B, 5C and 5D). Among them, the first (anteriormost) supraorbital is the largest of the series, ornamented with strong ridges and tubercles. The second supraorbital is shorter

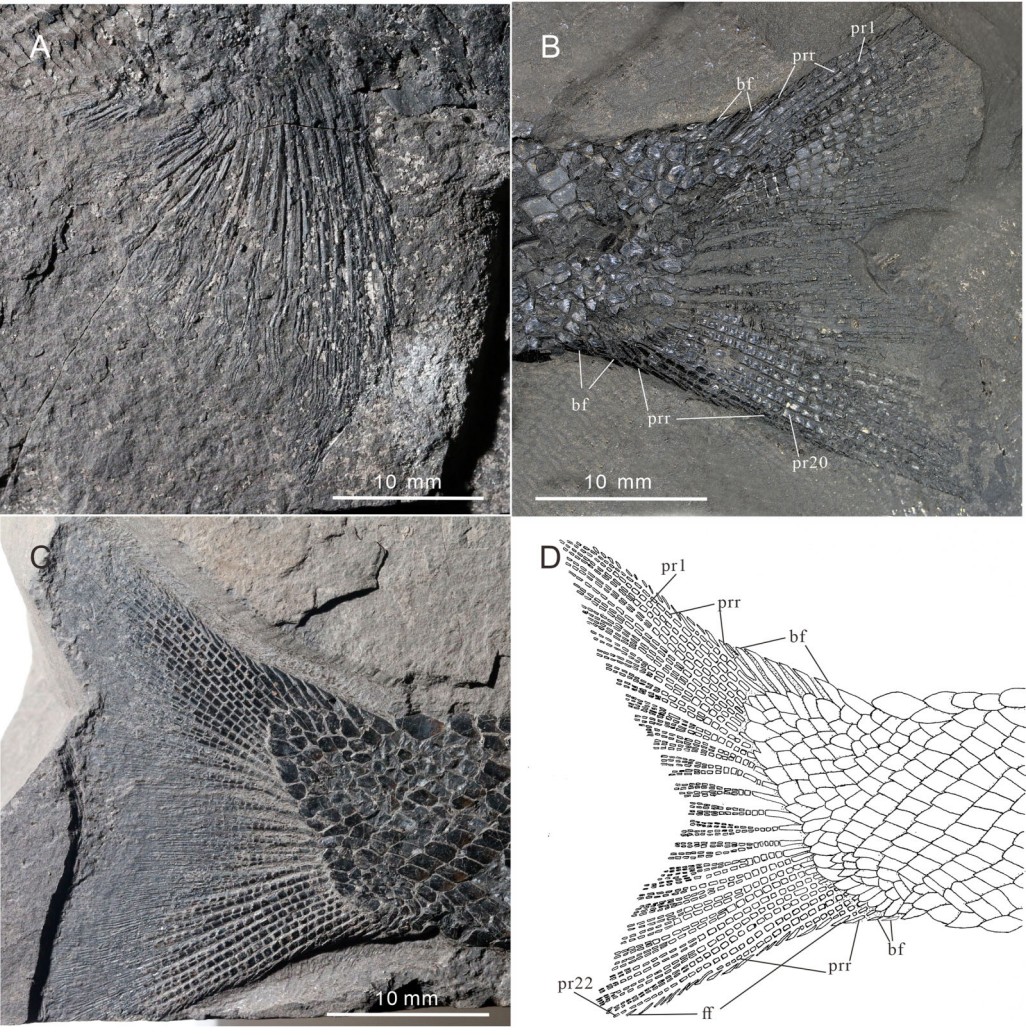

**Figure 9 Pectoral fin and caudal fin of *Teffichthys elegans* sp. nov.** (A) CUGM K2-E2606, pectoral fin. (B) CUGM K2-E2602, caudal fin. (C and D) CUGM K2-E2609; (C) caudal fin, (D) line-drawing of the caudal fin. All of them from the late Dienerian at Gujiao section.

and ornamented with weak tubercles. The third (posteriormost) supraorbital is nearly the same length as the second one, with weak ornamentation on the surface. It was split into two parts in the holotype (Fig. 5A).

Two infraorbitals, the lacrimal and the jugal, are present (Figs. 4A–4D, 5C and 5D). The lacrimal is elongate, dorsoventrally short, and slightly concave dorsally. It is not fused with the maxilla. The jugal is roughly crescent and expanded dorsally. It bears a prominent spiny process along its dorsal margin, which might be the channel of the infraorbital sensory canal. In addition, a tubercle occurs in the anterior portion of the jugal on both sides (Figs. 4A–4D).

The dermosphenotic is key-stone shaped and dorsally expanded, slightly deeper than wide (Figs. 4A–4D, 5C and 5D). The bone contacts the frontal and dermopterotic dorsally and the jugal ventrally. The infraorbital sensory canal extends through the entire

length of the infraorbital and dermosphenotic, then enters the dermopterotic posteriorly, finally connects with the supratemporal sensory canal.

**Cheek.** There are a suborbital and a spiracular (Figs. 4C and 4D). The suborbital is located posterior to the dermosphenotic. It is quadrangular in shape, slightly shorter than the dermosphenotic, and has a smooth posterior margin. The spiracular is smaller and subtriangular, contacting the suborbital anteriorly, the preopercle ventrally, and the dermopterotic dorsally.

The dermohyal bone is quadrangular, contacting the dermopterotic dorsally, and the preopercle ventrally (Figs. 4A–4D). It has a relatively large size compared with other Perleididae.

The postspiracular bone (Figs. 4C and 4D) is small, elongate, and nearly smooth, contacting the extrascapular dorsally, the dermohyal anteriorly, the opercle ventrally, and the posttemporal posteriorly. Notably, it is only exposed on the left side of the holotype.

The maxilla is anteriorly elongated and posteriorly expanded in shape, contacting the preopercle with a gently curved posterodorsal suture (Figs. 4A–4D, 5C and 5D). It bears a row of approximately 25 conical teeth along the slightly curved oral margin, and the size of the teeth decreases posteriorly. The lateral surface of the maxilla is ornamented with dense tubercles and ridges, although some of them were lost during specimen preparation.

**Palatal bones and suspensorium.** The hyomandibula is not exposed. The rounded molariform teeth of the palatal bones are present, exposed next to the posterior-dorsal portion of the maxilla (Fig. 6D).

**Lower jaw.** The dentary is exposed on both sides of the holotype. It is slender and elongated with a nearly straight oral margin and a convex ventral margin, posteriorly bounded by a small angular (Figs. 4A and 4B). The dentition along the oral margin of the dentary consists of more than 24 peg-like teeth, and their length decreases posteriorly. The supra-angular and coronoid process are not discernible. The angular is small and irregular in lateral view, tapering anteroventrally (Figs. 4A–4D). The dentary is ornamented with small tubercles and transverse running, interrupted ridges, and the latter are mainly distributed near the oral margin. The mandibular sensory canal crosses ventrally the entire length of the dentary and the angular.

**Operculogular series.** The opercle is subtriangular in shape, having a nearly straight margin in contact with the dermohyal anterodorsally (Figs. 4A–4D, 5C and 5D). The subopercle is trapezoidal in shape with a concave dorsal margin. It is larger than the opercle, possessing a moderately developed anterodorsal process (Figs. 4A–4D, 5C and 5D).

The preopercle is large and broad, contacting the second infraorbital (jugal) and spiracular anteriorly, the dermopterotic dorsally, and the dermohyal and opercle posteriorly (Figs. 4A–4D, 5C and 5D). The ventral portion of the preopercle is wedged between the maxilla and subopercle. The dorsal portion of the preopercle tapers dorsally,

having a narrow contact with the dermopterotic. The preopercular sensory canal extends along the posterior margin of the preopercle, indicated by a series of small pores (Figs. 4 and 5).

Six pairs of branchiostegal rays are present below the dentary and angular (Figs. 4A and 4B). Four pairs are mediolaterally elongated and nearly equal in size in the anterior region. The latter two pairs broaden posteriorly, having the same width as the anterior branchiostegal rays. A large and nearly oval bone, anterior to branchiostegals, is interpreted as the median gular (Fig. 6A). It shows a tapered anterior margin and convex posterior margin. The median gular is approximately one-third the length of the dentary. Only the right lateral gular is preserved. It is trapezoidal in shape and overlapped by the first branchiostegal ray in the posterior portion (Figs. 5A and 5B).

**Pectoral girdle and paired fins.** The posttemporals are subrectangular in shape, and widely separated from each other by the mid-dorsal scales (Figs. 4E and 4F). The posttemporal contacts the dermopterotic anteriorly, the opercle laterally, and the supracleithrum posteriorly. The supracleithrum is a large bone, deep and quadrangular in shape, and inclined forward (Figs. 4A–4D). It is ornamented with dense tubercles and ridges mainly on the middle and anterior portion of this bone. There are two vertically arranged postcleithrums below the supracleithrum, showing dense tubercles on the surface (Figs. 4A–4D). The dorsal postcleithrum is rectangular in shape, with its dorsal process overlapped by the supracleithrum. The ventral postcleithrum is irregular in shape, with a concave dorsal margin, and it is smaller than the dorsal postcleithrum. The cleithrum is elongated, curved, and inclined backwards (Figs. 4A–4D, 5C and 5D). It is anteriorly overlapped by the subopercle, and mainly ornamented with ridges on the surface.

The pectoral fins insert low on the body. The exoskeletal part of the pectoral fin is large, nearly the same size as the anal fin (Figs. 2 and 9A). Sixteen distally segmented and twice branched rays are present in the pectoral fin in CUGM K2-E2606 (Fig. 9A). The first ray is preceded by more than 10 fringing fulcra and two basal fulcra. In the holotype, there are at least five short, rod-like radials between the cleithrum and the exoskeletal portion of the pectoral fin (Figs. 4A, 4B and 8D). The endoskeletal part of the pectoral fin is not exposed.

The pelvic fin originates at the 14th vertical scale row. It is smaller than the pectoral fin, and contains six distally segmented fin rays, preceded by two basal fulcra and at least 13 fringing fulcra (Fig. 8C). The first pelvic ray is unbranched, and the remaining five rays are branched twice distally.

**Median fins.** The dorsal fin originates above the 26th vertical scale row (Fig. 8B). It is composed of three procurrent rays and 12 principal rays, preceded by three basal fulcra and at least 16 fringing fulcra. All rays are segmented distally. The procurrent rays become longer posteriorly, and the third procurrent ray is the longest one, being approximately three times the length of the first principal ray. The first principal ray, slightly shorter than the second, is unbranched, and the other rays are branched twice distally. The second principal ray is the longest ray of the dorsal fin, and other principal rays become shorter

posteriorly. There are seven exposed thick, rod-like pterygiophores supporting the dorsal fin. Each of them nearly corresponds to a single fin ray, indicating that rays and pterygiophores are nearly equal in number (Fig. 8B).

The anal fin originates below the origin of the 22th vertical scale row (Fig. 8A). It contains two procurrent rays and 9 principal rays. The fin contains two unbranched rays, preceded by two basal fulcra. All rays are segmented distally, and only the first three rays are unbranched. Both the procurrent rays are associated with fringing fulcra along their anterior margin. Notably, the first procurrent ray is very short, nearly one-third of the second one. The endoskeleton of the anal fin is not exposed.

The caudal fin, well preserved in specimens CUGM K2-E2607–2608 (Figs. 3A, 3B and 9B–9D), is abbreviated heterocercal (backbone only extending slightly into upper lobe of the caudal fin, *Thomas, Bonner & Whiteside (2007)* with a forked posterior profile. The caudal fin consists of 20–23 principal rays. There are three procurrent rays and seven epaxial basal fulcra in the dorsal lobe (Fig. 4B), and six procurrent rays and two basal fulcra in the ventral lobe (Figs. 9C and 9D). The surfaces of the rays are ornamented with slender tubercles, parallel to the direction of fin rays, generally one or two in number (Figs. 9C and 9D).

**Squamation.** The body is entirely covered by ganoid scales. The scales are arranged in 39–41 vertical rows along the lateral line (Figs. 2 and 3). The lateral line scales are the deepest scales in each vertical row. Each of them has a small notch at its posterior margin, and two small pores are present in the anterior portion in some lateral line scales. The lateral line scales are deep in the anterior trunk region. They decrease gradually in size toward the posterior region and become rhomboidal on the caudal peduncle. In the anterior region of the trunk, the scales are deep and narrow. They have numerous well-developed denticles at the posterior margin, and tubercles and ridges are well developed on the free field. In contrast, in the posterior flank region, the scales are rhomboidal and smooth, and the denticles at the posterior margin gradually decrease in number posteriorly.

### Character comparisons

*Teffichthys elegans* sp. nov. possesses diagnostic features of *Teffichthys* (*Marramà et al., 2017*), presence of suborbital and spiracular ossicles, the posttemporals widely separated, maxilla fixed to the slightly forward inclined preopercle, opercle smaller than subopercle, presence of fringing fulcra on all fins, and presence of an abbreviated heterocercal caudal fin. Additionally, the numbers of supraorbitals and branchiostegal rays of the new species are also within the range of *Teffichthys*. However, it differs from *T. madagascariensis* in four features (Fig. 10A; Table 1). First, six pairs of branchiostegal rays appear in *T. elegans* sp. nov., while *T. madagascariensis* has four to five. Second, developed spines are present at the posterior margin of the jugal in *T. madagascariensis*, but they are absent in *T. elegans* sp. nov. Third, four supraorbitals are present in *T. madagascariensis*, while three supraorbitals occur in *T. elegans* sp. nov. Fourth, presence

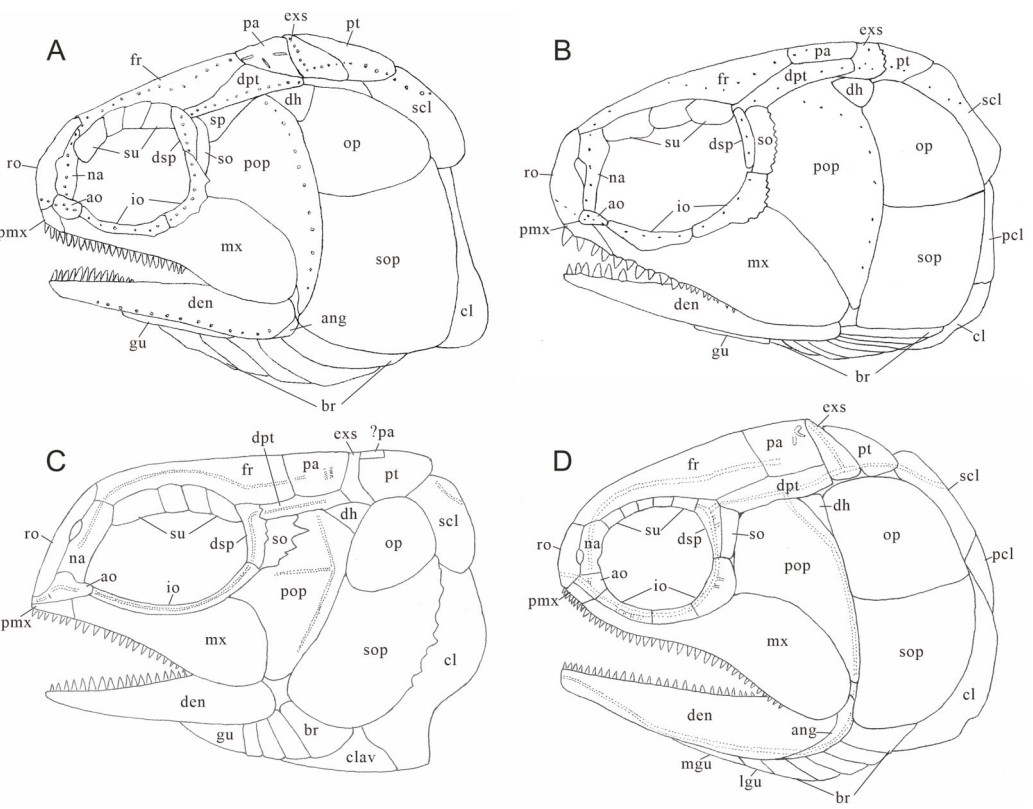

**Figure 10 Comparison of skull and pectoral girdle in Perleididae.** (A) *Teffichthys madagascariensis* (adapted from *Marramà et al., 2017*); (B) *Perleidus altolepis* (adapted from *Lombardo, 2001*); (C) *Meidiichthys browni* (adapted from *Hutchinson, 1973*); (D) *Plesiofuro mingshuica* (adapted from Xu, 2015).

**Table 1 Summary of characters used to differ members of Perleididae.**

| Species | Br | Su | Io | So | Pt/exs | Psp | Ppr | Na contact | La fused with mx | Formula | SL/mm |
|---|---|---|---|---|---|---|---|---|---|---|---|
| *Perleidus altolepis* | 6 | 3 | 2 | 1 | 1/2 | Absent | Present | Absent | Absent | D19/P12A19C35/T37 (PH) | <120 |
| ***Teffichthys elegans* sp. nov.** | 6 | 3 | 2 | 2 | 1/2 | Present | Present | Absent | Absent | D26/P14A22C36/T39 | >148 |
| *Teffichthys madagascariensis* | 4–5 | 4 | 2 | 2 | 1/2 | Absent | Present | Absent | Absent | D25/P13A21C37/T40 | 51–129 |
| *Paraperleidus changxingensis* | ? | 3 | 2 | 1 | 1 | Absent | Present | Present | Absent | ? | 265 (TL) |
| *Plesioperleidus yangtzensis* | 6 | 3 | 1 | 1 | 1 | Absent | Present | Absent | Present | D30/P15A25C46/T51 | 200 (TL) |
| *Plesiofuro mingshuica* | 5 | 5 | 3 | 1 | 1 | Absent | Absent | Absent | Absent | D28/P14A25C39/T44 | 90–120 |
| *Meidiichthys browni* | 4 | 4 | 1 | 1 | 1 | Absent | Absent | Absent | Absent | D21/P13A21C35/T38 (PH) | 65–100 (TL) |

Note:
Data from *Brough (1931)*, *Lombardo (2001)*, *Jin, Wang & Cai (2003)*, *Zhao & Lu (2007)*, Xu (2015), *Marramà et al. (2017)*. *Paraperleidus* and *Plesioperleidus* are based on personal observation (GH Xu and ZW Yuan, 2021). TL, total length; PH, based on reconstruction picture.

of a relatively deep anterodorsal process of subopercle in *T. elegans* sp. nov. *vs.* absence in *T. madagascariensis*.

    *Teffichthys elegans* sp. nov. can be distinguished from *"Perleidus" woodwardi* in two features. First, unlike *T. elegans* sp. nov., the dermopterotic is nearly the same length of the parietal in *"Perleidus" woodwardi* (*Stensiö, 1921*, Plate 34). Second, *T. elegans* sp. nov. has

small and conical teeth on the posterior oral margins of both jaws and relatively weak crushing teeth on the palatal bones; by contrast, *"Perleidus" woodwardi* has blunt teeth on the oral margin of posterior portions of both jaws and robust crushing teeth on the pterygoid, indicating a more powerful durophagous ability than that in the former.

*Teffichthys elegans* sp. nov. can be distinguished from *"Perleidus" stoschiensis* in three features. First, the posttemporal is nearly the same length of the extrascapular in *"Perleidus" stoschiensis* (*Stensiö, 1932*, Plate XXI., Figs. 3 and 4), while the posttemporal is nearly half the length of the extrascapular in *T. elegans* sp. nov. Second, five supraorbitals are present in *"Perleidus" stoschiensis* (*vs.* three in *T. elegans* sp. nov.). Third, the preopercle has a broad dorsal portion in *"Perleidus" stoschiensis*, but the bone tapers dorsally into a pointed tip in *T. elegans* sp. nov.

The major difference between *Teffichthys elegans* sp. nov. and *Perleidus* are the cheek bone, the teeth on the jaws, and the caudal fin (Fig. 10B). *T. elegans* sp. nov. has a posteriorly smooth suborbital and a spiracular, leading to a thin connection between the preopercle and dermopterotic. However, *Perleidus* possesses a posteriorly serrated suborbital, leading to a broad contact region between the preopercle and dermopterotic. Additionally, the teeth of *Perleidus* are stronger than those of *T. elegans* sp. nov. Moreover, *Perleidus* has a nearly symmetrical abbreviated heterocercal caudal fin, possessing epaxial principal rays and about five epaxial procurrent rays (*Lombardo, 2001*; *Lombardo et al., 2011*); while *T. elegans* sp. nov. has only three epaxial procurrent rays and no epaxial principal rays, forming a typical abbreviated heterocercal caudal fin. Unlike *Plesiofuro* and *Meidiichthys*, *Teffichthys elegans* sp. nov. has two infraorbitals (three in *Plesiofuro*, one in *Meidiichthys*; Figs. 10C and 10D), a spiracular (absent in *Plesiofuro* and *Meidiichthys*), six branchiostegals (five in *Plesiofuro*, four in *Meidiichthys*), and three epaxial procurrent rays in the caudal fin (absence of epaxial procurrent rays in *Plesiofuro* and *Meidiichthys*; Table 1).

*Teffichthys elegans* sp. nov. can be distinguished from *Paraperleidus* by the following features: presence of a spiracular, the width of the posttemporal being half of the width of the extrascapular, and the anterior portions of nasals separated by rostral (Figs. S1A and S1B; Table 1). In addition, the standard length of *T. elegans* sp. nov. is approximately 148 mm, much smaller than *Paraperleidus*, which reaches 265 mm in standard length (*Zhao & Lu, 2007*). Notably, *Paraperleidus* differs from other perleidids in three features. First, the nasal bones are joined in the midline and located posterior to the rostral. In stem-neopterygians, this feature is otherwise present in Platysiagiformes and *Thoracopterus* within Peltopleuriformes (*Griffith, 1977*; *Tintori & Sassi, 1992*; *Wen et al., 2019*). Second, the width of posttemporal is nearly the same as the width of extrascapular. This feature is otherwise present in *Meidiichthys*, *Plesiofuro*, and *Plesioperleidus* within Perleididae (*Brough, 1931*; *Hutchinson, 1973*; *Jin, Wang & Cai, 2003*; *Tong et al., 2006*; *Xu, Zhao & Shen, 2015b*). Third, it has less than three epaxial procurrent rays in the caudal fin, showing a condition different from those in *Perleidus*, *Plesiofuro*, and *Meidiichthys* (Fig. S2D).

*Teffichthys elegans* sp. nov. differs from *Plesioperleidus* in four features: width of posttemporal being only half of the width of extrascapular, absence of fused lacrimo-maxilla, absence of an anteriorly extended portion of preopercle, and presence of a

spiracular (Figs. S1C, S1D; S2A–S2C, S2E; Table 1; *Jin, Wang & Cai, 2003*; *Tong et al., 2006*). In addition, *Plesioperleidus* differs from other perleidids in the following features: width of posttemporal being nearly the same as the width of extrascapular, similar to *Paraperleidus*; fusion of the lacrimal with the maxilla occurred in some specimen (*e.g.*, IVPP V5343, CUGM J2203a; Fig. S2E), similar to the conditions in *Feroxichthys* and Luganoiiformes (*Bürgin, 1992*; *Xu, 2020b*; *Xu, 2020c*; *Ma, Xu & Geng, 2021*); presence of an anteriorly extended ventral portion of preopercle, which is absent in other Perleididae; and presence of three epaxial procurrent rays in the caudal fin, which is absent in *Plesiofuro* and *Meidiichthys*. (Fig. S2B).

## Phylogenetic affinities

The phylogenetic analysis results in 54 most parsimonious trees (tree length = 484 steps, consistency index = 0.401, retention index = 0.752), the strict consensus of which is presented in Fig. 11. In the cladogram, Perleididae is recovered as a stem lineage of crown-neopterygians, nesting between Colobodontidae and Louwoichthyiformes.

The Perleididae shares the following derived features of Colobodontidae and more derived neopterygians (Fig. S4): presence of molariform teeth on the oral bones (absent in *Plesiofuro*, Louwoichthyidae, *Altisolepis*, *Venusichthys*, and more derived neopterygians), width of posttemporals being about half width of extrascapular series (reversal in *Plesiofuro*, *Meidiichthys*, *Paraperleidus*, *Plesioperleidus*, and some crown-neopterygians), quadratojugal much reduced or lost (splint-like in *Lepisosteus*, *Semionotus*, and *Kyphosichthys*), equal number of dorsal and anal fin rays relative to pterygiophores, and dorsal and anal fin rays only being segmented distally. The sister group relationships between Perleididae and more derived neopterygians are supported by the following features (Fig. S4): presence of no more than six pairs of branchiostegals (reversal in *Thoracopterus* and some crown-neopterygians), and 24 or fewer principal rays in both lobes of the caudal fin (reversal in *Ctenognathichthys*, *Luopingichthys*, *Fuyuanperleidus*, and *Caturus*). Perleididae lacks the derived features of Louwoichthyiformes and more derived neopterygians: the ratio of dermopterotic length to parietal length was less than two (reversal in *Dipteronotus*, *Moradebrichthys*, and some crown-neopterygians), and the anteroventral corner of the preopercle is lower than the ventral end of the opercle (absent in Peltopleuriformes, *Habroichthys*, and crown-neopterygians).

In Perleididae, *Meidiichthys* and *Plesiofuro* form a sister group due to the absence of the epaxial procurrent rays (*Hutchinson, 1973*; Xu, 2015). *Paraperleidus*, *Plesioperleidus*, and *Meidiichthys-Plesiofuro* clade form a polytomy clade, supported by the following feature: width of the posttemporal is nearly as wide as the extrascapular (*Hutchinson, 1973*; *Jin, Wang & Cai, 2003*; *Tong et al., 2006*; *Zhao & Lu, 2007*; Xu, 2015). *Teffichthys elegans* sp. nov. and *Teffichthys madagascariensis* form a sister group due to the presence of suborbital and spiracular ossicles. The sister relationship between the *Teffichthys* clade and the polytomy clade is supported due to the presence of less than three epaxial procurrent rays and absence of epaxial principal rays (*Marramà et al., 2017*). The former is more derived than the latter due to the width of posttemporal nearly half of the width of the extrascapular. *Perleidus* is recovered as the basalmost taxon within the Perleididae by the

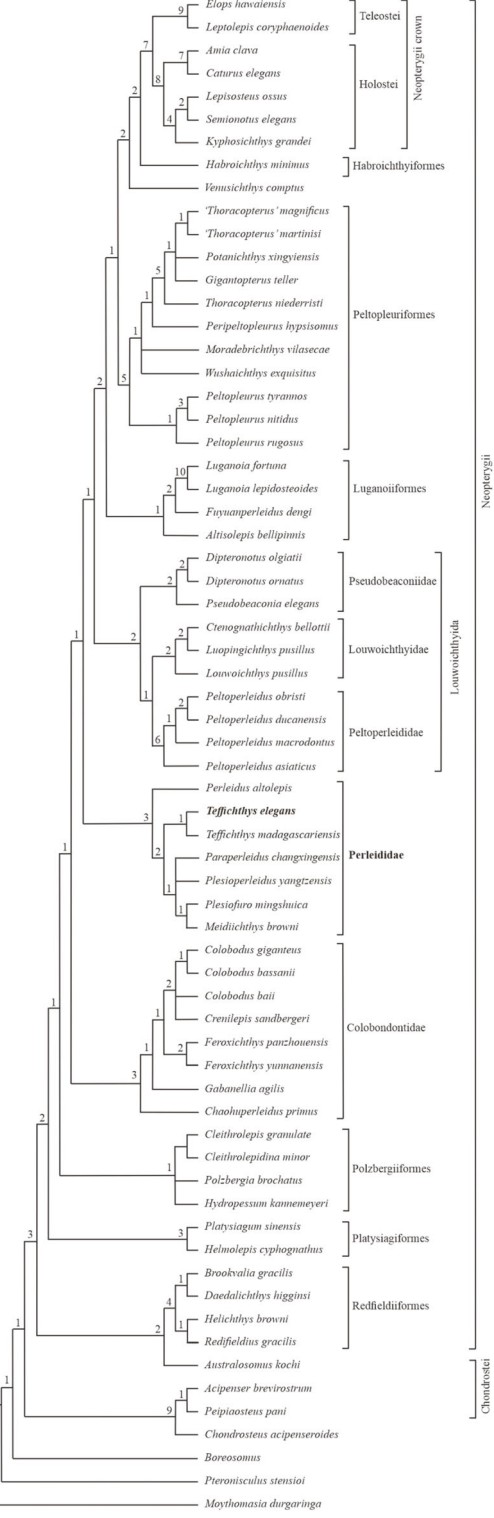

**Figure 11 Strict consensus of phylogenetic analysis of Perleididae and relative Actinopterygii.** Strict consensus of 54 most parsimonious trees (tree length = 484 steps, consistency index = 0.401, retention index = 0.752), illustrating the phylogenetic position of *Teffichthys elegans* sp. nov. within the Actinopterygii. Numbers above nodes indicate Bremer decay indices.

following feature: presence of more than three epaxial procurrent rays and presence of epaxial principal rays (*Lombardo, 2001*; *Lombardo et al., 2011*).

Notably, two genera, previously referred to the Perleididae, are herein reallocated within other families. *Chaohuperleidus* is recovered as a member of Colobodontidae due to the presence of two synapomorphies of this family: a prominent anterodorsal process of the subopercle and presence of rounded ganoid tubercles on the principal caudal fin rays (*Sun et al., 2013*, Fig. 5). *Moradebrichthys* is herein recovered at the base of Thoracopteridae, and forms the sister group to *Wushaichthys* and more derived taxa of this family (*Xu, Zhao & Shen, 2015b*; *Cartanyà et al., 2019*). It possesses several synapomorphies of Thoracopteridae: laterally expanded frontals, parietal fused with dermopterotic, and posttemporal contacting extrascapular posterolaterally and separating this bone from contact with its counterpart.

Two families, Gabanellidae and Hydropessidae, are excluded from the Perleidiformes. *Gabanellia*, previously included within Gabanellidae, is recovered as a member of the Colobodontidae by the following features: a prominent anteroventral extension of subopercle, the length of the anteroventral margin of the preopercle (Fig. S3; where the preopercle contact with the maxilla) nearly same as the length of the anterodorsal margin of the preopercle (*Tintori & Lombardo, 1996*). *Hydropessum*, previously considered as a member of the Hydropessidae, is recovered within Polzbergiiformes due to the absence of teeth on the maxilla and the posterior end of the maxilla ending below the posterior orbital margin (*Broom, 1909*; *Hutchinson, 1973*).

In addition, results of our analysis provide new insights into the phylogenetic relationships of some stem-neopterygian taxa. *Redfieldius* and *Helichthys* are recovered as sister taxa due to the absence of dermopterotic/preopercle contact (*Hutchinson, 1973*; *Schaeffer & McDonald, 1978*). *Peltopleurus tyrannos* forms the sister taxon to *Peltopleurus nitidus* due to the presence of following features: anal fin being larger than dorsal fin, presence of a single suborbital; presence of a broad suborbital posterior to the dermosphenotic, with the same depth with each other; absence of epaxial principal rays in the caudal fin; and the length of the anteroventral margin of the preopercle nearly the same as the length of the anterior margin of the subopercle (*Xu & Ma, 2016*; *Xu, Ma & Zhao, 2018*).

## DISCUSSION

### The monophyly of Perleididae and Perleidiformes

The monophyly of Perleididae can be supported by three synapomorphies: the length of the anteroventral margin of the dermohyal (Fig. S3) nearly half of the length of the anterodorsal margin of the preopercle (unknown in *Paraperleidus* due to poor preservation; present in *Australosomus* and *Platysiagum*); the anteroventral margin of the preopercle is nearly equal to the anterior margin of the subopercle in length (independently evolved in Platysiagiformes, Luganoiiformes, and *Peltopleurus*); and the anteroventral margin of the preopercle is one to two times as long as anterodorsal margin of the preopercle (also present in *Australosomus* and *Colobodus*). *Procheirichthys*, *Aetheodontus*, *Meridensia*, *Alvinia*, *Megaperleidus*, *Endennia*, *Chaohuperleidus*, and

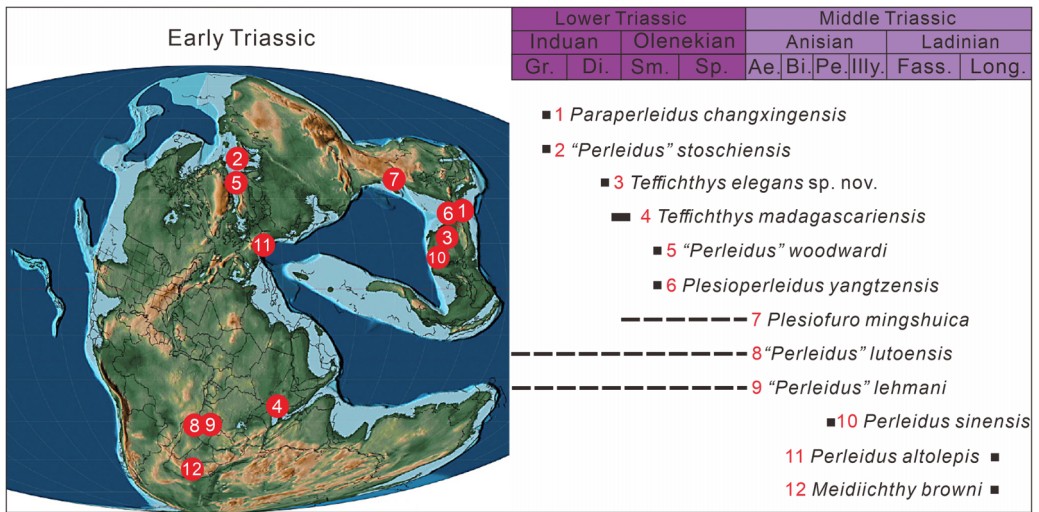

**Figure 12 The paleogeographic distribution of Perleididae in Early and Middle Triassic, modified from (*Scotese, 2014*).** Abbreviated stratigraphic intervals: Gr, Griesbachian; Di, Dienerian; Sm, Smithian; Sp, Spathian; Ae, Aegean; Bi, Bithynian; Pe, Pelsonian; Illy, Illyrian; Fass, Fassanian; Long, Longobardian.                         

*Moradebrichthys* are moved out of the Perleididae as they lack the three synapomorphies listed above (*Wade, 1935*; *Bürgin, 1992*; *Sytchevskaya, 1999*; *Lombardo & Brambillasca, 2005*; *Sun et al., 2013*; *Cartanyà et al., 2019*). Additionally, the taxonomic positions of *Diandongperleidus* and *Luopingperleidus* are ambiguous due to the poor preservation and require further studies. At present, the monophyly of Perleididae is still weakly supported, pending further works in the future.

We propose that the Perleididae should be considered as the only valid family of the Perleidiformes. Hydropessidae and Gabanellidae were once referred to the Perleidiformes. However, *Hydropessum* shows many features resembling Polzbergiiformes, such as the absence of teeth on the maxilla and the posterior end of the maxilla ending below the posterior orbital margin (*Broom, 1909*; *Hutchinson, 1973*). Additionally, *Gabanellia* possesses several features which are strikingly different from the Perleididae: presence of a modified anal fin, presence of more than ten branchiostegal rays, and presence of powerful teeth of different sizes (*Tintori & Lombardo, 1996*). As such, Hydropessidae and Gabanellidae are removed from the Perleidiformes to keep the monophyly of this order.

## Biogeographical and ecological implications

Perleididae (as redefined herein) reached a cosmopolitan distribution in the late Induan and early Olenekian. Similar cosmopolitanism has been also documented in ammonoids, terrestrial tetrapods, and some actinopterygian fishes (*e.g.*, Saurichthyidae, Birgeriidae, Parasemionotidae) during the Early Triassic (*Brayard et al., 2006*; *Romano et al., 2016a*; *Button et al., 2017*; *Dai & Song, 2020*). Following this, Perleididae was restricted to the eastern rim of the Paleotethys Ocean in the Anisian and appeared at its western rim in the Ladinian (Fig. 12). In the Induan, Perleididae occurred in South China, East Greenland, and Madagascar (*Stensiö, 1932*; *Lehman, 1952*; *Zhao & Lu, 2007*; *Marramà*

*et al., 2017*). At the beginning of the Olenekian, Perleididae underwent rapid radiation. The family occurred in Spitsbergen, South China, and Madagascar during the Smithian, demonstrating a global distribution pattern (*Woodward, 1912*; *Stensiö, 1921*; *Lehman, 1952*; *Jin, Wang & Cai, 2003*; *Marramà et al., 2017*). With the removal of *Chaohuperleidus* from Perleidiformes, there is no record of Perleididae in the Spathian, when actinopterygian fishes were rarely reported (*Romano, 2021*). In addition, there are some taxa of Perleididae from North China and Angola that lack specific ages at the stage/substage level and require further examination (*Antunes et al., 1990*; *Xu, Gao & Coates, 2015a*). In the Middle Triassic, Perleididae was discovered in South China in the Anisian, showing its latest record from southern Europe in the Ladinian (*Brough, 1931*; *Hutchinson, 1973*; *Lombardo, 2001*; *Lombardo et al., 2011*).

The discovery of *Teffichthys elegans* sp. nov. in Guizhou documents the first record of neopterygians in the Upper Yangtze region of South China. Previously, Early Triassic fish fossil localities in China mainly focused on the Middle and Lower Yangtze region, namely Jurong, Jiangsu (*Qian, Zhu & Zhao, 1997*; *Liu et al., 2002*; *Jin, Wang & Cai, 2003*), Chaohu, Anhui (*Tong et al., 2006*), and Huangshi, Hubei (*Su & Li, 1983*). However, the new species described here comes from eastern Guizhou in the Upper Yangtze region.

Having a stand length of 148 mm, *Teffichthys elegans* sp. nov. was one of the largest durophagous predators of Perleididae in the Early Triassic. Similar to other species of *Teffichthys* and *Perleidus*, *T. elegans* sp. nov. possessed a streamlined body and a large orbital region, providing the advantage of energy efficiency for relatively fast swimming and good visual acuity (*Lombardo, 2001*; *Marramà et al., 2017*). In addition, the new species possesses a feeding apparatus similar to other perleidids in having a long maxilla with a triangular postorbital portion, a relatively stout dentary, conical teeth occupying most of the oral margins of both jaws, and molariform teeth on the palatal bones. The molariform teeth in *T. elegans* sp. nov., although less developed than those in the Colobodontidae (a group of large durophagous predatory stem-neopterygian fishes; *Bürgin, Arratia & Viohl, 1996*; *Mutter, 2004*; *Xu, 2020a*), would enable it to feed on some hard-shelled organisms, such as crustaceans, bivalves, and ammonoids, which also yielded from the same horizon at Gujiao section (*Dai et al., 2018*). *T. elegans* sp. nov. likely represents a secondary consumer resembling other perleidids (*Scheyer et al., 2014*). The new discovery may indicate a relatively complex trophic structure of the Early Triassic marine ecosystem in South China.

## CONCLUSIONS

The discovery of *Teffichthys elegans* sp. nov. from the Induan (Dienerian), Early Triassic of Guizhou, South China, sheds light on the radiation of stem-neopterygians of China after the Permian/Triassic mass extinction. The results of this phylogenetic analysis recovered *T. elegans* sp. nov. as a member of Perleididae (as a monophyletic group redefined herein), sister to *T. madagascariensis*, and provided new insights into the relationship of early neopterygian clades. *Chaohuperleidus* and *Moradebrichthys* were moved to Colobodontidae and Thoracopteridae, respectively. The Perleididae is considered here as the only valid family of the Perleidiformes; Hydropessidae and

Gabanellidae should be removed from the Perleidiformes to maintain the monophyly of this order. *T. elegans* sp. nov. is one of the largest durophagous predators of perleidids in having a combination of molariform and conical teeth. The new finding may indicate a relatively complex trophic structure of the Early Triassic marine ecosystem in South China.

## ACKNOWLEDGEMENTS

We thank Toni Bürgin, Carlo Romano, Giuseppe Marramà, and Kenneth De Baets for helpful comments on an early version of the manuscript. We thank Lijun Zhao for the specimen observation of *Paraperleidus* and *Plesioperleidus* in Zhejiang Museum of Natural History, Hangzhou, China. We thank the National infrastructure of Mineral Rock and Fossil Resources for Science and Technology of Yifu Museum of China University of Geoscience for the management of the specimens. We thank Yiran Cao, Xin Yang, Shuxun Yuan, and Yi Wang for their assistance in the fieldwork.

### Funding

This study was supported by the National Natural Science Foundation of China (92155201, 41821001, 42172008), the Strategic Priority Research Program of Chinese Academy of Sciences (XDB26000000), and the 111 Project (B08030). This study is a contribution to IGCP 630. The funders had no role in study design, data collection and analysis, decision to publish, or preparation of the manuscript.

### Grant Disclosures

The following grant information was disclosed by the authors:
National Natural Science Foundation of China: 92155201, 41821001 and 42172008.
Strategic Priority Research Program of Chinese Academy of Sciences: XDB26000000.
111 Project: B08030.

### Competing Interests

Haijun Song is an Academic Editor for PeerJ.

### Author Contributions

- Zhiwei Yuan conceived and designed the experiments, performed the experiments, analyzed the data, prepared figures and/or tables, authored or reviewed drafts of the paper, for the obtain of materials, and approved the final draft.
- Guang-Hui Xu conceived and designed the experiments, analyzed the data, authored or reviewed drafts of the paper, and approved the final draft.
- Xu Dai performed the experiments, prepared figures and/or tables, authored or reviewed drafts of the paper, for the obtain of materials, and approved the final draft.
- Fengyu Wang performed the experiments, authored or reviewed drafts of the paper, for the obtain of materials, and approved the final draft.
- Xiaokang Liu performed the experiments, authored or reviewed drafts of the paper, for the obtain of materials, and approved the final draft.

- Enhao Jia performed the experiments, authored or reviewed drafts of the paper, for the obtain of materials, and approved the final draft.
- Luyi Miao performed the experiments, authored or reviewed drafts of the paper, for the obtain of materials, and approved the final draft.
- Haijun Song conceived and designed the experiments, performed the experiments, analyzed the data, prepared figures and/or tables, authored or reviewed drafts of the paper, and approved the final draft.

## Data Availability

The supplementary figure, character list, and data matrix are available in the Supplemental File.

## New Species Registration

The following information was supplied regarding the registration of a newly described species:

Publication LSID: urn:lsid:zoobank.org:pub:D32FE268-726E-47C6-A2D9-F669836F3424

Genus *Guiyangichthys* gen. nov.

LSID: urn:lsid:zoobank.org:act:D516B686-825D-4990-9929-9996D6C971A6

Genus *Teffichthys* LSID: urn:lsid:zoobank.org:act:FE3CD0A3-B197-49C0-94B1-4929AF2025F9

Species *Guiyangichthys elegans* sp. nov.

LSID: urn:lsid:zoobank.org:act:7AD93E0E-25F6-415E-8BBD-43872084F3D8.

## Supplemental Information

Supplemental information for this article can be found online at http://dx.doi.org/10.7717/peerj.13448#supplemental-information.

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
