# Peer review of "A new perleidid neopterygian fish from the Early Triassic (Dienerian, Induan) of South China, with a reassessment of the relationships of Perleidiformes"

_PeerJ, doi:10.7717/peerj.13448_

## Round 0.1 · original submission · Major Revisions

The manuscript describes a new species of perleidid fish based on well preserved material. The new taxon and supporting phylogenetic analysis have great potential and merit to improve our understanding of early perleidids and the evolution of neopterygian fishes after the Permo-Triassic extinction. Some (syn)apomorphies and characters need to be reconsidered and re-analyzed. The main points which need to be addressed are:

1) The erection of a new genus: All characters used to the define fall within the morphological and meristic range of Teffichthys based on articulated material from Early Triassic of Madagascar by Marramà et al., 2017 (see reviewer 2). The erection a new genus Guiyangichthys is not necessary nor justified. The combination of characters, however, differs (enough) from that of other Teffichthys species recognized so far justifying the erection of a new species (T. elegans) of Teffichthys (see reviewer 2).

2) The presence of a postspiracle: The designation of this bone as an automorphy of the new taxon seems to be erratic and if it is truly the first appearance in the perleididae, it needs to be shown in the reconstruction of Figure 7 which lacks the spiracle bones (compare reviewer 1 and 2). More importantly, the position of this bone designated in the literature under various names (spiracle, postspiracle, spiracularis or spiracular) has also been used to diagnose the genus Teffichthys and likely present in other paleoniscoid fishes although it is unclear if it is homologous (compare reviewer 2). Either way, it is not the first time that it appeared in Perleididae as it was already described in Teffichthys and should be coded as present in this genus in the phylogenetic analysis. Various other characters also speak for the identification of your material as new species of Teffichthys including presence of three supraorbitals; presence of two infraorbitals; six pairs of branchiostegals rays; relative deep anterodorsal process of subopercle; the lacrimal separated from maxilla; the nasals don’t contact with each other; the width of posttemporal is half of the width of extrascapular; presence of epaxial procurrent rays; pterygial formula of D26/P14, A22, C36/T39-41 in addition to the presence of postspiracle (see reviewer 2). Please considering to first describe the specimens and characterize the species before analyzing its phylogenetic placement (compare reviewer 3).

3) The definition and diagnosis of the Perleididae: the monophyly of the Perleididae is currently based on synapomorphies of difficult value (compare reviewer 1). As pointed out by reviewer 2, an emended diagnosis of the family is necessary because the two characters (the preopercle process length of dermohyal nearly being half the length of the anterodorsal process of the preopercle, and the anteroventral corner of the preopercle being nearly the same horizon as the ventral end of opercle) are just those supporting the monophyly of the family in the phylogenetic analysis.

4) Presence of epaxial rays: procurrent rays should not be confused with principal epaxial caudal-fin rays (see Lombardo 2001 and other references listed in the review by reviewer 2). This also was not picked up by the reviewers in the previous study by Ma et al. (2021) because Teffichthys HAS NO epaxial principal caudal-fin rays. For these reasons as nicely detailed by reviewer 2, statements and coding in the matrix for characters 112 and 113 must be changed for the new taxon, Teffichthtys madagascariensis, Plesioperleidus yangtzensis and for all the other taxa in which you considered these procurrent rays as “epaxial rays.

5) Removal of Hydropessidae and Gabanellidae from the order Perleidiformes: I agree with reviewer 2 that you cannot remove both families without testing their phylogenetic relationships. As you did not include Hydropessum nor Gabanellia, your statements are currently not supported by data or analysis. As opposed to your statements, well-preserved and articulated material is available for both Gabanellia (Tintori & Lombardo 1996) as well as Hydropessum (Hutchinson 1973; reviewer 2 list the full references). I therefore recommend including these two taxa into a new phylogenetic analysis to see the validity of your statements rather than the alternative of removing these statements concerning these taxa (compare reviewer 2).

6) Phylogenetic analysis: reviewer 2 obtained a less resolved strict consensus tree with TNT used the parameters stated in the Material & Methods section and could only reproduce the tree of Figure 10 using a 50% Majority Rule and not using a Strict consensus. Please verify and re-analyze. The same reviewer also obtained the same topology with a sister-group relationship between "Guiyangichthys" elegans and other species of Teffichthys using different states for epaxial rays and postspiracular. I agree with reviewer 2 that including a new character might give you further grouping information: the caudal fin type: [0] true heterocercal (e.g., paleonisciforms’); [1] abbreviated heterocercal (e.g., Teffichthys); [2] symmetrical heterocercal (e.g., Perleidus); [3] homocercal (Teleostei). These as well as other points raised previously (e.g., possible inclusion of Hydropessum and Gabanellia, recoding epaxial rays as procurrent rays, presence of postspiracle) merit a phylogenetic re-analysis.

7) Comparative analysis: please consider including comparative material (e.g., figures) in the main text or supplementary material to support your arguments (e.g., for presence or rather the lack of epaxial rays in Teffichthys). Morphological comparisons between Guiyangichthys, Paraperleidus, etc. might also benefit from adding a figure with their reconstructed skulls (or whole skeleton), and highlighting differences and similarities (compare reviewer 3). As the holotype is nicely preserved in 3D, it might be worth to do a CT scan for this or a future study. Either way a statement concerning the potential for CT might be beneficial (compare reviewer 3)

8) Formatting and language: the manuscript is mostly well written and understandable, but formatting and language can be further improved (see reviewer 3). Some sentences in the description are quite long and might benefit from breaking them up in smaller sentences. I am not a native speaker myself but agree with reviewer 3 that having a colleague fluent in English to proofread the manuscript before resubmission would be beneficial.

9) Bibliography: please make sure all citations in text are cited in the reference list and vice versa (compare reviewer 3)

10) Palaeogeographic maps: Reviewer 3 highlighted some errors in your maps, please make sure these are addressed.

11) Figure captions: Figure captions should be expanded with all necessary information about each specimen. Figures with labelled skeletal parts should ideally contain the legend in the caption, which makes it easier to read them (compare reviewer 3)

Please make sure these and all other points raised by the reviewers including those listed in annotated pdfs are addressed. I feel most if not all these points are minor to moderate and easy to address, particularly with the constructive suggestions made by the reviewers. However, as reviewers pointed out issues which call for re-assessment of some characters and phylogenetic re-analysis, I feel another round of reviews is necessary. Hence my decision for major rather than minor revisions.

Thank you for your understanding and I look forward to receiving the revised manuscript.

·

Basic reporting

The manuscript in review is a sound and concise description of a new perleidid species. It is based on well preserved material. I have made some minor corrections in the manuscript.

The presence of a postspiracle (267ff) seems to be somewhat erratic. If it is the first appearance in the perleididae, it should be shown in the reconstruction (fig. 7).

389-390: Peltopleurus nitidus formed a sister group with Peltopleurus nitidus …should be Peltopleurus tyrannos

396ff: the monophyly of the Perleididae is based on two synapomorphies of difficult value: in my opinion the mentioned character states are somewhat shaky due to morphological plasticity (see presence/absence of postspiracle).

Experimental design

The phylogenetic analysis seems to be consistent, but see my comment on above mentioned synapomorphies.

Validity of the findings

The drawings and illustrations are well done and support the descriptive part, as well as the conclusions.

Additional comments

The description of this new perleidid species provides new and interesting data on this important fossil fish family and is a valuable contribution to our understanding of it.

Reviewer 2 ·

Basic reporting

The article is well written, clear, unambiguous, technically correct. Literature and background context are correctly provided. Article structure, figures and tables are correctly provided and shared. The manuscript is an important study about a new taxon from the Early Triassic of South China that might have the potential to improve our understanding on the evolution of neopterygian fishes after the Permo-Triassic extinction. However, the manuscript is affected by several issues that must be necessarily considered before re-submitting the manuscript. Based on comments below and in the attached pdf, I suggest MAJOR REVISIONS.

Experimental design

This is an original research, questions are well defined. However, the research lacks of rigorous investigation and several statements are unsupported. See below.

Validity of the findings

The manuscript is an important study about a new taxon from the Early Triassic of South China that might have the potential to improve our understanding on the evolution of neopterygian fishes after the Permo-Triassic extinction. However, the manuscript is affected by several issues that must be necessarily considered before re-submitting the manuscript. Based on comments below and in the attached pdf, I suggest MAJOR REVISIONS.

1) The authors report the following combination of characters to diagnose the new genus (and species): A new species of Perleididae distinguished from other members of this family by the following features (autapomorphies identified with an asterisk): presence of postspiracle (*); presence of three supraorbitals; presence of two infraorbitals; six pairs of branchiostegals rays; relative deep anterodorsal process of subopercle; the lacrimal separated from maxilla; the nasals don’t contact with each other; the width of posttemporal is half of the width of extrascapular; presence of epaxial procurrent rays; pterygial formula of D26/P14, A22, C36/T39-41 (*).
Although the new taxon described likely represent a new species, the combination of characters used to diagnose Guiyangichthys elegans coincides exactly (or is within the morphological and meristic ranges) with that of the genus Teffichthys Marramà et al., 2017, created on articulated material from Early Triassic of Madagascar. In particular:
• Spiracular ossicles are diagnostic of Teffichthys (Marramà et al. 2017), so it cannot be considered as an autapomorphy of this new taxon. These bones are present in all the other early Triassic Teffichthys species, and also in some "paleoniscoid" fishes (although their homology must be still detected). The authors repeatedly ignore in the text that this bone (spiracle, postspiracle, spiracularis, or spiracular) has been used to diagnose Teffichthys, so it is not the first time it appeared in Perleididae (if also Teffichthys must be considered a Perleididae). This bone is indicated in literature with several (similar) names, but since the position of this bone is the same in Peltopleuriformes, Teffichthys, and the taxon described herein, it is likely these bones are homologous in all these taxa. Conversely, it is unclear if the spiracle of paleoniscoids is homologous. The presence of this bone in Teffichthys must be therefore coded in the phylogenetic analysis as present.
• Also the presence of three supraorbitals; two infraorbitals; six pairs of branchiostegals rays (used to diagnose Guiyangichthys, are within the range of the genus Teffichthys !!! although these numbers vary among the other Early Triassic species (T. lehmani, T. lutoensis, T. stoschiensis, T. woodwardi). Variation among the different species occurs also in the squamation formulae.
• The same can be said for many other features used to diagnose Teffichthys and characterizing “Guiyangichthys” elegans, including: a body elongate and tapered; dermal cranial bones ornamented with tubercles or ridges; parietal pit-lines (anterior, median and posterior); posttemporals separated; suborbital and spiracular ossicles present; two to five supraorbitals; nasals separated by the rostral; maxilla fixed to an almost vertical preopercle; nearly straight oral margin of maxilla, which is dorso-posteriorly expanded; jaws with styliform teeth; broad vertical or slightly forward inclined preopercle; opercle smaller than subopercle; five to eight branchiostegal rays; dorsal and anal fins inserted in the posterior half of the body; median-fin rays only distally segmented and supported by an equal number of pterygiophores at least in the middle part; fringing fulcra present on all fins; abbreviated heterocercal caudal fin; anteriormost lateral trunk scales higher than wide, with serrated posterior margin.
• For these reasons, the creation of a new genus is unsupported, being all these characters within morphological and meristic ranges of Teffichthys. However, since the combination of meristic and morphological character is different from those of the other Teffichthys species recognized so far, the new taxon can be considered a new species of Teffichthys (T. elegans).

2) Another main problem is the question about the epaxial rays. This terminology has been clarified several times in the literature, but the authors have ignored it. “Epaxial rays" are PRINCIPAL caudal-fin rays (not procurrent rays!! ) positioned dorsal to the notocord (or to the upper caudal fin). This is the original definition sensu Hutchinson 1973; Gardiner & Schaffer 1989; Gardiner 1988; Tintori 1990 (see also Lombardo 2001 for references), that several authors continue to misinterpret or ignore since Stansio (1929). Lombardo (2001) and subsequent authors have clarified this concept, but several authors still misinterpret EPAXIAL PRINCIPAL CAUDAL-FIN RAYS with epaxial procurrent rays.
In this perspective, based on the original definition of “epaxial caudal fin rays”, the new taxon described herein HAS NO epaxial PRINCIPAL caudal fin rays (but just some procurrent rays) dorsal to the upper caudal fin lobe. Therefore, the statement of Ma et al (2021) reported by the authors at lines 99-100 must be considered wrong because Teffichthys HAS NO epaxial principal caudal- fin rays. This would even better support the relationship between T. madagascariensis and the new species described herein.
For these reasons, statements and coding in the matrix for characters 112 and 113 must be changed for the new taxon, Teffichthtys madagascariensis, Plesioperleidus yangtzensis and for all the other taxa in which the authors considered these procurrent rays as “epaxial rays”.

3) The authors cannot remove Hydropessidae and Gabanellidae from the order Perleidiformes without testing their phylogenetic relationships. The phylogenetic analysis the authors performed does not include Hydropessum nor Gabanellia. For this reason, their statements at lines 30-33, 411-413, and 507-508, are not supported by data. Remove the statements or include these two taxa in a new phylogenetic analysis to see the validity of the statements.

Moreover, the authors stated that these two taxa cannot be included in their phylogeney because of the poorly preserved material. However, Gabanellia has been erected by Tintori & Lombardo (1996) based on well preserved articulated and nearly complete material. A complete description of Hydropessum with high quality images was provided by Hutchinson (1973). Therefore their statement about the poorly preserved material is not supported. Again, please include Gabanellia and Hydropessum, and test their relationships, or remove these statements.
See:
Hutchinson 1973. A revision of the redfieldiiform and perleidiform fishes from the Triassic of bekkers kraal south africa and brookvale new-south-wales.
Tintori & Lombardo 1996) Gabanellia agilis gen. n. sp. n., (Actinopterygii, Perleidiformes) from the Calcare Di Zorzino of Lombardy (North Italy). Riv It Paleontol Stratigr 742: 227-236.

4) I would suggest at this point to provide an emended diagnosis of the family Perleididae, because the two characters (the preopercle process length of dermohyal nearly being half the length of the anterodorsal process of the preopercle, and the anteroventral corner of the preopercle being nearly the same horizon as the ventral end of opercle) are just those supporting the monophyly of the family in the phylogenetic analysis. An emendation of the diagnosis of the family is therefore needed.

5) PHYLOGENETIC ANALYSIS. Using the same parameters stated in Material & Methods with TNT I got a less resolved Strict consensus tree than that depicted in Figure 10. A similar tree to that in Figure 10 can be obtained only as 50% Majority Rule tree, or, alternatively, without saving multiple trees (multiple trees saved OFF). In any case, I got the same tree topology using different states for Teffichthys (epaxial rays absent, and postspiracular present). Teffichthys is still sister to "Guiyangichthys" elegans therefore suggesting that the absence of epaxial rays in Teffichthys does not affect the tree topology.

I would also suggest to include a new character that might give you further grouping information: the caudal fin type: [0] true heterocercal (e.g. paleonisciforms’); [1] abbreviated heterocercal (e.g. Teffichthys); [2] symmetrical heterocercal (e.g. Perleidus); [3] homocercal (Teleostei). ( you might try also to group states 1 and 2 in a single state)

6) The reconstruction of the skull at Figure 7 does not show the spiracular bones. Please indicate these bones.

Additional comments

I hope that the authors will consider these suggestions. I look forward to see a revised version of the paper.

Annotated reviews are not available for download in order to protect the identity of reviewers who chose to remain anonymous.

·

Basic reporting

Although the ms is mostly well written and understandable, there are several language errors (choice of word, sentence structure, missing or superfluous words, past/present tense, punctuation). I highlighted them in the annotated pdf. Authors must do a language check by a native speaker before resubmission.

In addition, some sentences in the description are not fully clear. Authors must ensure that their major findings are clear and understandable. Several sentences are very long and is is difficult to follow. Authors should try to shorten sentences and make sure their main points are easily understood by readers.

Some citations in the text lack references. Some references are not cited in the text. Authors must check their references carefully before resubmission.

The order of subchapters could be improved. I think after the description a thorough discussion of the morphology of Guiyangichthys is most logical, including a comparison with other perleidiforms and early neopterygians. After that, the phylogenetic analysis should follow. This would be more logical as the authors must first clearly establish the validity of their new taxon before analysing its phylogenetic position. Morphological comparisons between Guiyangichthys, Paraperleidus, etc. can be better supported by adding a figure with their reconstructed skulls (or whole skeleton), and highlighting differences and similarities.

The palaeogeographic maps contain some errors, which need to be addressed. Figure captions are very brief. They should be expanded with all necessary information about each specimen. Figures with labelled skeletal parts should ideally contain the legend in the caption, which makes it easier to read them.

Methods section should include comparative material. For comparisons, authors could also consider publishing figures of comparative specimens to support their arguments (i.e. epaxial rays of Teffichthys).

Since the holotype is preserved in 3D, did the authors consider to ct scan the specimen? The new taxon is mainly described based on external features. With ct data, the endocranial morphology could be described, and the importance of the paper could be further increased. If ct scans of the specimen proved not useful, authors should mention this in their ms.

Regarding other comments, please check the annotated pdf.

Experimental design

No further comment here. The description should partly be expanded to better characterize this new taxon.

Validity of the findings

This is a new taxon of perleidid fish, a group of high interest for the study of early neopterygian phylogeny. The material is well-preserved and adds to our knowledge of ealry perleidids. The analyses are sound.

---

## Round 0.2 · Minor Revisions

Thank you for addressing the reviewers’ suggestions so thoroughly which makes the manuscript for consistent and easier to follow – your manuscript is as good as accepted. There are just some minor issues which still need to be addressed:

• Terminology: please use the appropriate terminology for phylogenetic analyses (see reviewer 2)

• Caudal fin pattern: please discuss the differences in caudal fins patterns between Perleidus and Teffichthys in greater detail (compare reviewer 2)

• References: Please make sure that in all paragraphs, appropriate sources are cited/referenced when discussing similarities and differences between taxa (see reviewer 3)

• Formatting/Language issues: please make sure technical terms and grammar/syntax is correct (see reviewers 2 and 3)

• Figure caption: please make sure the figure captions (e.g., Fig. 10) are consistent with the labels in the figure (see reviewer 1)

Please make sure these as well as other points raised by reviewers including those in annotated pdfs are addressed. I look forward to obtaining the revised manuscript and seeing this work published.

·

Basic reporting

Due to the substantial and thorough remarks of the two other reviewers the paper of Yuan et al. improved substantially. I have read their responses to the reviewers comments and it’s in my opinion, that the paper can be published now.

Experimental design

no comment

Validity of the findings

see 1.

Additional comments

On the second version of the manuscript I have only one comment: On figure 10 the drawings of the four different head reconstructions should be labeled according to the text of the figure: A, B, C, D.

Reviewer 2 ·

Basic reporting

I really appreciated that my suggestions from the first review were taken into account. The manuscript is now almost ready to be accepted. Only minor revisions are still needed:
1) Use the correct terminology for phylogenetic analyses. There is no "at the top" or "at the bottom" in a phylogenetic tree ! You may say Perleidus is recovered sister to... or as the basalmost taxon within the Perleididae but there is no top, bottom, further up, further down...Please use correct terminology.
2) In character comparisons and throughout the ms the caudal fin pattern of Perleidus and Teffichthys needs to be discussed in more detail, as it is very different in the two genera. Please compare the caudal fins described in Lombardo (2001) and Marramà et al. (2017).
3) A further proofcheck is needed, as the technical terminology must be enanched, and grammar or synthax can be improved. Some suggestions and corrections are included within the pdf attached but please check again introduction and description.

Dr. Giuseppe Marramà, PhD

Experimental design

No other comments needed.

Validity of the findings

Findings and the whole research are now valid.

Additional comments

No other comments needed.

Annotated reviews are not available for download in order to protect the identity of reviewers who chose to remain anonymous.

·

Basic reporting

The manuscript has improved after the first round of review.

There are a few language errors (highlighted in the annotated pdf).

Some paragraphs are poorly referrenced. Authors must cite sources when discussing similarities or differences between T. elegans and other taxa, or refer to the Supplementary File if applicable. See annotated PDF. In addition to the differences of T. elegans with other closely related taxa, authors should also briefly discuss similarities with T. elegans, as well as Teffichthys/Perleidus stoschiensis and Teffichthys/Perleidus woodwardi. This is important to better support the validity of the Chinese species. Figure 10 looks nice!

Experimental design

No further comments necessary

Validity of the findings

No new comments necessary

Additional comments

No further comments.

---

## Round 0.3 · Minor Revisions

Thank you for addressing the suggestions of the reviewers. Reviewer 2 pointed out some additional points concerning the terminology of suborbitals, spiracular and postspiracular as well as the captions of Figure 10 which I would like to see resolved before publication. I look forward to receiving the revised version and the publication of this manuscript.

Reviewer 2 ·

Basic reporting

The terminology of some bones, particularly the suborbitals, spiracular and postspiraculars, needs to be better addressed. The authors indicate two suborbitals in Teffichthys, but Marramà et al. indicate one suborbital and one spiracular (this latter seems identify as suborbital by the Yuan et al. Please clarify this and check all descriptions of these bones and reconstruction figures.
Moreover, you have to modify the caption for Figure 10, because A is Teffichthys madagascariensis, not Meidiichthys brown (which is C). Moreover, the spiracular of Teffichthys madagascariensis is indicated as suborbital. Please check carefully all bone abbreviations.
Once the authors solved these minor revisions the paper can be published as it is.

Experimental design

OK

Validity of the findings

OK

·

Basic reporting

No further comments.

Experimental design

No further comments.

Validity of the findings

No further comments.

Additional comments

I am satisfied with the additions and corrections.

---

## Round 0.4 · accepted · Accept

Thank you for addressing the final suggestions by reviewer 2. As you appropriately resolved the issues highlighted by the reviewer, there is no need for further review. I look forward to seeing this manuscript published.